

# Lower rotational inertia and larger leg muscles indicate more rapid turns in tyrannosaurids than in other large theropods

Eric Snively[1], Haley O'Brien[2], Donald M. Henderson[3], Heinrich Mallison[4], Lara A. Surring[3], Michael E. Burns[5], Thomas R. Holtz Jr[6,7], Anthony P. Russell[8], Lawrence M. Witmer[9], Philip J. Currie[10], Scott A. Hartman[11] and John R. Cotton[12]

[1] Department of Biology, University of Wisconsin-La Crosse, La Crosse, WI, USA
[2] Department of Anatomy and Cell Biology, Oklahoma State University Center for Health Sciences, Tulsa, OK, USA
[3] Royal Tyrrell Museum of Palaeontology, Drumheller, AB, Canada
[4] Museum fur Naturkunde, Berlin, Germany
[5] Department of Biology, Jacksonville State University, Jacksonville, AL, USA
[6] Department of Geology, University of Maryland, College Park, MD, USA
[7] Department of Paleobiology, National Museum of Natural History, Washington, D.C., USA
[8] Department of Biological Sciences, University of Calgary, Calgary, AL, Canada
[9] Department of Biomedical Sciences, Ohio University, Athens, OH, USA
[10] Department of Biological Sciences, University of Alberta, Edmonton, AL, Canada
[11] Department of Geoscience, University of Wisconsin-Madison, Madison, WI, USA
[12] Department of Mechanical Engineering, Ohio University, Athens, OH, USA

Corresponding author
Eric Snively, esnively@uwlax.edu

## ABSTRACT

**Synopsis:** Tyrannosaurid dinosaurs had large preserved leg muscle attachments and low rotational inertia relative to their body mass, indicating that they could turn more quickly than other large theropods.

**Methods:** To compare turning capability in theropods, we regressed agility estimates against body mass, incorporating superellipse-based modeled mass, centers of mass, and rotational inertia (mass moment of inertia). Muscle force relative to body mass is a direct correlate of agility in humans, and torque gives potential angular acceleration. Agility scores therefore include rotational inertia values divided by proxies for (1) muscle force (ilium area and estimates of m. caudofemoralis longus cross-section), and (2) musculoskeletal torque. Phylogenetic ANCOVA (phylANCOVA) allow assessment of differences in agility between tyrannosaurids and non-tyrannosaurid theropods (accounting for both ontogeny and phylogeny). We applied conditional error probabilities $a(p)$ to stringently test the null hypothesis of equal agility.

**Results:** Tyrannosaurids consistently have agility index magnitudes twice those of allosauroids and some other theropods of equivalent mass, turning the body with both legs planted or pivoting over a stance leg. PhylANCOVA demonstrates definitively greater agilities in tyrannosaurids, and phylogeny explains nearly all covariance. Mass property results are consistent with those of other studies based on skeletal mounts, and between different figure-based methods (our main mathematical slicing procedures, lofted 3D computer models, and simplified graphical double integration).
**Implications:** The capacity for relatively rapid turns in tyrannosaurids is ecologically intriguing in light of their monopolization of large (>400 kg), toothed dinosaurian predator niches in their habitats.

## INTRODUCTION

Tyrannosaurid theropods were ecologically unusual dinosaurs (*Brusatte et al., 2010*), and were as adults the only toothed terrestrial carnivores larger than 60 kg (*Farlow & Holtz, 2002*) across much of the northern continents in the late Cretaceous. They ranged in adult trophic morphology from slender-snouted animals such as *Qianzhousaurus sinensis* (*Li et al., 2009*; *Lü et al., 2014*) to giant bone-crushers including *Tyrannosaurus rex* (*Rayfield, 2004*; *Hurum & Sabath, 2003*; *Snively, Henderson & Phillips, 2006*; *Brusatte et al., 2010*; *Hone et al., 2011*; *Bates & Falkingham, 2012*; *Gignac & Erickson, 2017*). In addition to the derived features of their feeding apparatus, the arctometatarsalian foot of tyrannosaurids likely contributed to effective prey capture through rapid linear locomotion and enhanced capability of the foot to resist torsion when maneuvering (*Holtz, 1995*; *Snively & Russell, 2003*; *Shychoski, Snively & Burns, 2011*). Features suggestive of enhanced agility (rate of turn) and tight maneuverability (radius of turn) in tyrannosaurids include relatively short bodies from nose to tail (anteroposteriorly short thoracic regions, and cervical vertebrae that aligned into posterodorsally retracted necks), small forelimbs, and long, tall ilia for leg muscle attachment (*Paul, 1988*; *Henderson & Snively, 2003*; *Bakker & Bir, 2004*; *Hutchinson et al., 2011*). Here, we present a biomechanical model that suggests tyrannosaurids could turn with greater agility, thus pivoting more quickly, than other large theropods, suggesting enhanced ability to pursue and subdue prey.

Like other terrestrial animals, large theropods would turn by applying torques (cross products of muscle forces and moment arms) to impart angular acceleration to their bodies. This angular acceleration can be calculated as musculoskeletal torque divided by the body's mass moment of inertia (=rotational inertia). Terrestrial vertebrates such as cheetahs can induce a tight turn by lateroflexing and twisting one part of their axial skeleton, such as the tail, and then rapidly counterbending with the remainder, which pivots and tilts the body (*Wilson et al., 2013*; *Patel & Braae, 2014*; *Patel et al., 2016*). The limbs can then accelerate the body in a new direction (*Wilson et al., 2013*). These tetrapods can also cause a larger-radius turn by accelerating the body more quickly with one leg than the other (pushing off with more force on the outside of a turn), which can incorporate hip and knee extensor muscles originating from the ilium and tail (Table 1). Hence muscles originating from the ilium can cause yaw (lateral pivoting) of the entire body, although they do not induce yaw directly. Such turning balances magnitudes of velocity and lean angle, and centripetal and centrifugal limb-ground

**Table 1 Muscles originating from the ilium and tail of theropod dinosaurs (*Carrano & Hutchinson, 2002*; *Mallison, Pittman & Schwarz, 2015*) and their utility for yaw (turning the body laterally).**

| Muscle | Action | Effect on turning (yaw) |
|---|---|---|
| **Ilium origin** | | |
| M. iliotibialis 1 | Knee extension, hip flexion | Greater acceleration outside turn, stabilization inside turn |
| M. iliotibialis 2 | Knee extension, hip flexion | Greater acceleration outside turn, stabilization inside turn |
| M. iliotibialis 3 | Knee extension | Greater acceleration outside turn, stabilization inside turn |
| M. iliotrochantericus caudalis | Hip abduction | Joint stabilization |
| M. iliofemoralis externus | Hip abduction | Joint stabilization |
| M. iliofemoralis internus | Hip abduction | Joint stabilization |
| M. caudofemorais brevis | Femoral retraction, direct yaw of body, pitch of body | Yaw with unilateral contraction, contralateral braking |
| **Tail origin** | | |
| M. caudofemoralis longus | Femoral retraction, direct yaw of body, pitch of body | Yaw with unilateral contractionIpsilateral yaw by conservation of angular momentum, contralateral braking |
| **Ilium origin, tail insertion** | | |
| M. ilio-ischiocaudalis (dorsal) | Tail lateral and dorsal flexion | Ipsilateral yaw by conservation of angular momentum, contralateral braking |

**Note:**
Although few muscles pivot the body directly over the stance leg (m. caudofemoralis brevis et longus, m. ilio-ischiocaudalis), all large ilium-based muscles are potentially involved with turning by acceleration of the body on the outside of the turn, stabilization of the hip joint, or conservation of angular momentum by swinging the tail.

forces. When limbs are planted on the ground, the body can pivot with locomotor muscle alone. In either case, limb muscles actuate and stabilize their joints, positively accelerating and braking the body and limbs.

Forces from locomotor muscles have a fundamental influence on agility. Torques from these limb muscles are necessary for estimating absolute angular acceleration (*Hutchinson, Ng-Thow-Hing & Anderson, 2007*), and muscle power also influences turning rate (*Young, James & Montgomery, 2002*). Experimental trials with human athletes show that agility scales with muscle force production relative to body mass (*Peterson, Alvar & Rhea, 2006*). Measures involving mass-specific force production, such as plyometric performance, correlate positively with agility tests of linear braking and accelerating (*Peterson, Alvar & Rhea, 2006*) and trials involving angular acceleration (*Thomas, French & Hayes, 2009*; *Markovic, 2007*; *Anderson et al., 1991*). Eccentric force, particularly of hamstrings in humans, is especially important for angular changes of direction (*Anderson et al., 1991*). Isolated indicators of maximal force of knee extensors alone contribute no more than 20% of variance to agility tests (*Markovic, 2007*), whereas functional, integrated force production and neurological training have larger effects on agility (*Peterson, Alvar & Rhea, 2006*; *Markovic, 2007*).

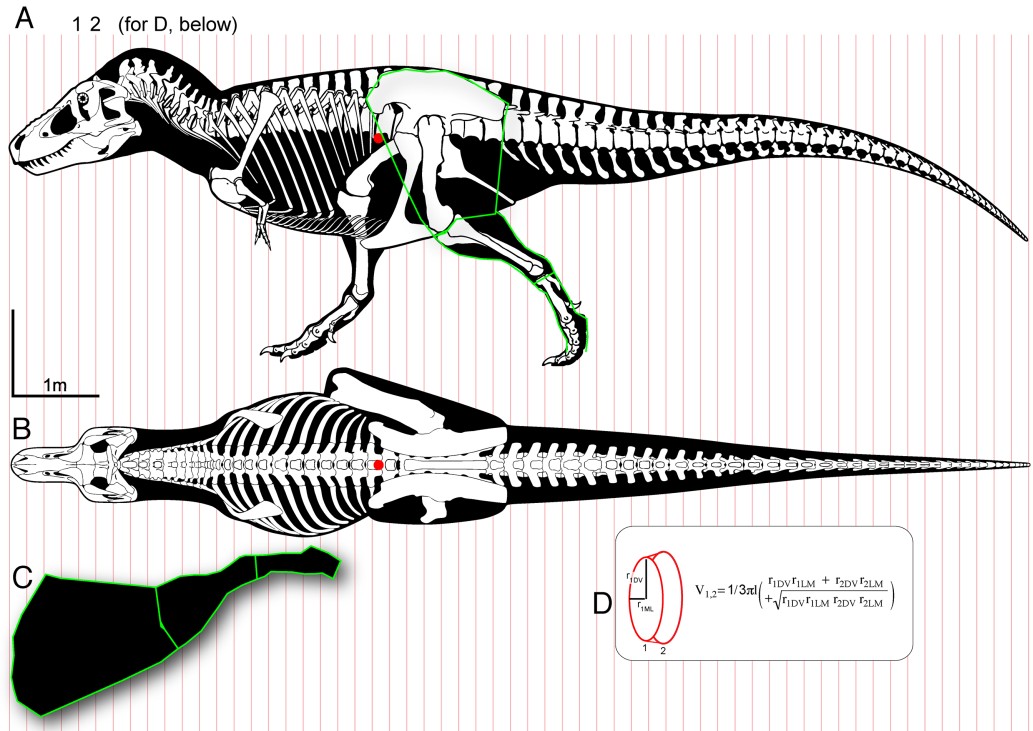

**Figure 1 Methods for digitizing body outlines and calculating mass properties, for "maximum tail width" estimate for *Tyrannosaurus rex*.** Reconstructions of *Tyrannosaurus rex* (Field Museum FMNH PR 2081) in lateral view (A) and dorsal view (B) enable digitizing of dorsal, ventral, and lateral extrema where they cross the vertical red lines. The lateral view (A) is modified with the dorsal margin of the neck conservatively raised based on recent muscle reconstructions (*Snively & Russell, 2007a*, *2007b*). The hind leg (A and C) is outlined in green, and straightened (C) for digitizing. A red dot (A and B) specifies the center of mass of the axial body (minus the limbs) using this reconstruction. An equation for the volume of a given frustum of the body (D), between positions 1 and 2, assumes elliptical cross-sections.

At a gross level (*Trinkaus et al. 1991*), muscle attachment size enables us to compare forces in fossil taxa, and to investigate relative agility. Muscle force is proportional to physiological cross-sectional area, and in turn on muscle volume, pennation angle, and dramatically on fiber length (*Bates & Falkingham, 2018*), in addition to maximal isometric stress and activation level. Muscle anatomical cross-sectional area and hence volume vary proportionally with attachment size of homologous muscles (explained in detail under Methods). In fossil taxa, attachment size is a consistent, reliably preserved influence on muscle force. Relative muscle force is therefore a useful, replicable metric for comparative assessments of agility in fossil tetrapods. Estimates of theropod muscle force and the mass properties of their bodies can facilitate comparisons of turning ability in theropods of similar body mass.

This relative agility in theropods is testable by regressing estimated body mass (Fig. 1) against indicators of agility, which incorporate fossil-based estimates of muscle force (Fig. 2), torque, and body mass and mass moment of inertia ($I_y$; Fig. 1). Given the same moment arm lengths, greater force relative to rotational inertia indicates the ability to turn more rapidly. Coupled with protracted juvenile growth periods
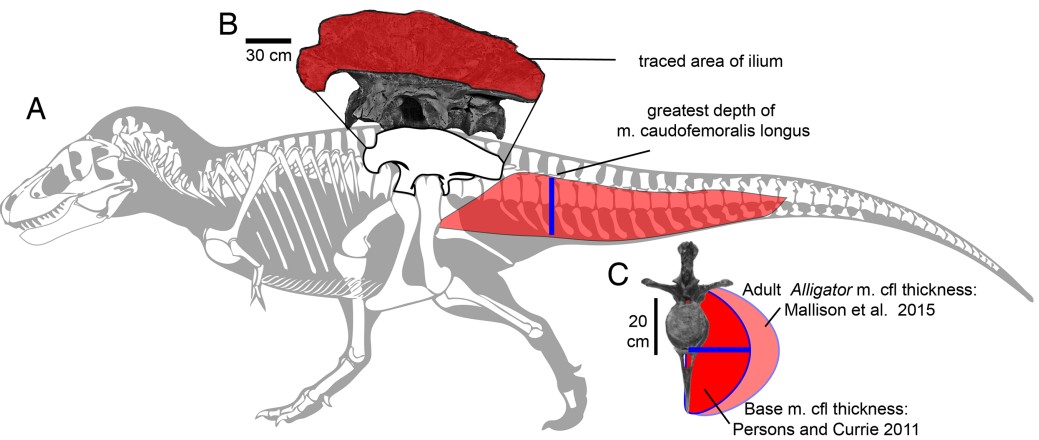

**Figure 2 Methods for approximating attachment cross-sectional area of hind limb muscles, on lateral view (A) of a *Tyrannosaurus rex* skeleton (FMNH PR 2081; modified from *Hartman, 2011*).** The blue line shows the position of the greatest depth from the caudal ribs to the ventral tips of the chevrons, and greatest inferred width of the m. caudofemoralis longus. (B) The inferred region of muscle attachment on the ilium (modified from *Brochu, 2003*) is outlined in red, for scaled area measurement in ImageJ. (C) The initial reconstructed radius (blue) of m. caufofemoralis longus (CFL) is 0.5 times the hypaxial depth of the tail (blue line in A), seen in anterior view of free caudal vertebra 3 and chevron 3. The maximum lateral extent of CFL is here based on cross-sections of adult *Alligator mississippiensis* (*Mallison, Pittman & Schwarz, 2015*). Note that the chevron in c is modified to be 0.93 of its full length, because it slopes posteroventrally when properly articulated (*Brochu, 2003*). Bone images in (A) and (C) are "cartoonized" in Adobe Photoshop to enhance edges.

(*Erickson et al., 2004*), heightened agility would be consistent with the hypothesis that tyrannosaurids were predominantly predatory, and help to explain how late Campanian and Maastrichtian tyrannosaurids monopolized the large predator niche in the Northern Hemisphere.

## Estimating mass properties and comparative turning performance of carnivorous dinosaurs

To compare agility in theropods, we divided ilium area (a proxy for muscle cross-sectional area and maximal force production), and estimated m. caudofemoralis longus (CFL) cross-sections, by $I_y$ (rotational inertia in yaw about the body's center of mass (COM)). We also incorporated scaling of moment arm size in a separate analysis to better compare absolute turning performance in the theropods. We restrict our comparisons to proxies of agility at given body masses, rather than estimating absolute performance, because a generalized predictive approach enables us to compare many taxa. Viable paths for testing our results include musculoskeletal dynamics of turning involving all hind limb muscles, as undertaken by *Rankin, Rubenson & Hutchinson (2016)* for linear locomotion in ostriches, or simpler approaches such as *Hutchinson, Ng-Thow-Hing & Anderson (2007)* calculations for turning in *Tyrannosaurus*. However, the dynamics of turning are complicated to pursue even in extant dinosaurs (*Jindrich et al., 2007*), and estimating absolute performance in multiple extinct taxa would entail escalating numbers of assumptions with minimal comparative return. We therefore focus here on relative metrics of turning performance, based as much as possible on direct fossil data.

Using relative indices of agility, encompassing origins for relevant ilium-based muscles, tail-originating muscles (Table 1), and mass moments of inertia, enables us to address action beyond yaw alone. Muscles of the leg on the outside of a turn normally involved in linear motion would change the body's direction by linearly accelerating the body in that direction, while muscles for the leg on the inside of the turn exert less torque. Muscles involved in stabilizing the limbs and body, and providing contralateral braking and abduction, would come into play during rotation of the body (Anderson et al., 1991). Mass moment of inertia is the most stringent mass-property limit on turning ability in long, massive dinosaurs (Carrier, Walter & Lee, 2001; Henderson & Snively, 2003). This simplified approach is predictive, testable with more complex investigations (including specific torques of muscle-bone couples: Hutchinson, Ng-Thow-Hing & Anderson, 2007), and allows broad comparisons of overall turning ability.

Our hypotheses of comparative agility in large theropods incorporate two behavioral scenarios potentially important for prey capture.

Hypothesis 1: Tyrannosaurids could turn their bodies more quickly than other theropods when close to prey, pivoting the body with both feet planted on the ground.
Hypothesis 2: Tyrannosaurids could turn more quickly than other theropods when approaching prey, pivoting the body plus a suspended swing leg above one stance foot planted on the ground.

Under the scenario in Hypothesis 1, the applicable mass moment of inertia $I_y$ is that of the body not including the hind legs, about a vertical axis through the body's COM. Intuitively the body would yaw about a vertical line between the acetabula, but the COM of bipedal dinosaurs, and therefore their feet and ground reaction forces in this stance, are almost always estimated to be anterior to the acetabulum (Henderson, 1999; Hutchinson, Ng-Thow-Hing & Anderson, 2007; Hutchinson et al., 2011; Allen, Paxton & Hutchinson, 2009; Allen et al., 2013; Bates et al., 2009a, 2009b; Bates, Benson & Falkingham, 2012).

In a prey pursuit scenario under Hypothesis 2, the theropod has just pushed off with its swing leg, and is pivoting about its stance leg as it protracts the swing leg. The body and swing leg are rotating about their collective COM, directly above the stance foot. Total $I_y$ in this case includes the entire axial body (minus the hind legs), and the contribution of the swing leg to total $I_y$ of the system.

## MATERIALS AND METHODS

Comparing relative turning performance in tyrannosaurids and other theropods requires data on mass moment of inertia $I_y$ about a vertical axis ($y$) through the body's COM, and estimates of leg muscle force and moment arms. $I$ in this paper always refers to mass moment of inertia, not $I$ as the common variable for area moment of inertia. To estimate mass, COM, and $I_y$, we approximated the bodies of the theropods as connected frusta (truncated cones or pyramids) with superellipse cross-sections (Fig. 1). Superellipses are symmetrical shapes the outline of which (from star-shaped, to ellipse, to rounded rectangle) are governed by exponents and major and minor dimensions (Rosin, 2000; Motani, 2001; Snively et al., 2013).

Spreadsheet templates for calculations of dimensions, mass, COM, and rotational inertias are available as Supplementary Information. These enable the estimation of mass properties from cross-sectional and length dimensions, using Microsoft Excel-compatible software. Snively et al. (2013) provide coefficients and polynomial regression equations for super-elliptical frusta.

## Specimens

Theropod specimens (Table 2) were included if they had complete ilia, and relatively complete skeletons ideally including the tail. If tails were incomplete they were reconstructed from other specimens of the same or a closely related genus, following the practice of Taylor (2009). Tyrannosaurid adults and juveniles are well represented by complete skeletons. Most other taxa were allosauroids, many of which are known from complete or rigorously reconstructable skeletons. *Yangchuanosaurus shangyouensis* and *Sinraptor hepingensis* are basal allosauroids. Their relative *S. dongi* lacks a preserved tail, and the older *Monolophosaurus jiangi* has a complete axial skeleton but lacks preserved hind legs, which are necessary for reliable mass estimates. Both species were therefore omitted. An early relative of allosauroids and tyrannosaurs, *Eustreptospondylus oxoniensis*, was included as a nearly complete, small representative of an allosauroid body plan, because it has a similar ratio of ilium/femur length as a less-complete juvenile specimen of *Allosaurus fragilis* (Foster & Chure, 2006), and is a reasonable proxy for the basal allosauroid condition. The non-tetanuran theropods *Dilophosaurus wetherelli* and *Ceratosaurus nasicornis* were included for their similarity in size to juvenile tyrannsaurids, and to enable examination of how phylogeny affects patterns of mass moment of inertia vs. muscle force. We include the small tyrannosaur that Sereno et al. (2009) named *Raptorex kriegsteini*. Fowler et al. (2011) provide evidence that this specimen is a juvenile *Tarbosaurus bataar* (see also Brusatte & Carr, 2016). We informally refer to it as *Raptorex* to differentiate it from a much larger juvenile *Tarbosaurus* in our sample.

## Digitizing of body outlines

Technical skeletal reconstructions by Paul (1988, 2010) and Hartman (2011), in dorsal and lateral views, were scanned on a flatbed scanner or saved as images (Hartman, 2011), vectorized with the Trace function in Adobe illustrator, and "expanded" for editing the entire outlines and individual bones. Lateral and dorsal outlines were modified based on body dimensions such as trunk, neck, and head length, and trunk and tail depth, as measured from scaled figures in the primary literature (Osborn, 1917; Gilmore, 1920; Russell, 1970; Dong, 1983; Gao, 1992; Brochu, 2003; Bates et al., 2009a, 2009b) and photographs of skeletons. We modified these outlines with updated anatomical data on neck and tail dimensions (Snively & Russell, 2007a; Allen, Paxton & Hutchinson, 2009; Persons & Currie, 2011a), and the jaws were positioned as closed. The chevrons of *Giganotosaurus* were angled posteroventrally to match those of its relatives *Acrocanthosaurus* and *Allosaurus*. Dorsal and lateral views were scaled to the same length, and divided into 60+ segments with lines crossing corresponding structures in both

**Table 2  Theropod taxa, specimens, and data sources for calculations of mass, mass moment of inertia, and ilium area.**

| Taxon | Specimen # | Lateral view | Dorsal view/modified from | Ilium source |
|---|---|---|---|---|
| *Dilophosaurus wetherelli* | UCMP 37302 | *Paul (2010)*, *Hartman (2015)*, and *Allen et al. (2013)* | *Paul (2010)*[†] and *Allen et al. (2013)* | *Hartman (2015)* |
| *Ceratosaurus nasicornis* | USNM 4735 | *Paul (2010)* | *Paul (2010)* | Photo; *Gilmore (1920)* |
| **Basal tetanurae** | | | | |
| *Eustreptospondylus oxoniensis* | OUM J13558 | *Paul (2010)* | *Paul (1988)* and *Walker (1964)* | *Walker (1964)* |
| *Allosaurus fragilis* | USNM 4734, UUVP 6000 | *Paul (1988, 2010)* | *Paul (2010)* | *Paul (2010)* and *Madsen (1976)* |
| *Allosaurus jimmadseni* (tail restored) | MOR 693 | *Bates et al. (2009a)* | *Paul (2010)* | Photo; *Loewen (2009)* |
| *Acrocanthosaurus atokensis* | NCSM 14345 | *Bates et al. (2009b)* | *Bates et al. (2009b)* | Photo, *Bates, Benson & Falkingham (2012)* (restored) |
| *Giganotosaurus carolinii* | MUCPv-CH-1 | *Paul (2010)* and *Hartman (2015)* | *Paul (2010)* and *Coria & Currie (2002)*[†] | Photo; *Hartman (2015)* |
| *Sinraptor hepingensis* | ZDM 0024 | *Paul (2010)* | *Paul (2010)* and *Gao (1992)* | *Gao (1992)* |
| *Yangchuanosaurus shangyouensis* | CV 00215 | *Paul (2010)* | *Paul (2010)* | *Dong, Zhou & Zhang (1983)* |
| **Tyrannosauroidea** | | | | |
| *Raptorex kriegsteini* (small juvenile *Tarbosaurus*) | LH PV18 | *Paul (2010)* | *Sereno et al. (2009)* | *Sereno et al. (2009)* |
| *Tarbosaurus bataar* (juvenile) | ZPAL MgD-I/3 | *Paul (1988, 2010)* | *Paul (1988)*[†] | Photo; *Paul (1988)* |
| *Tarbosaurus bataar* (adult) | ZPAL MgD-I/4 | *Paul (2010)* | *Hurum & Sabath (2003)* | Photo |
| *Tarbosaurus bataar* (adult) | PIN 552-1 | *Paul (2010)* | *Paul (1988)*[†] | *Paul (1988)* and *Maleev (1974)* |
| *Tyrannosaurus rex* (juvenile) | BMRP 2002.4.1 | *Paul (2010)* | *Persons & Currie (2011a)* | Photo; *Paul (2010)* |
| *Tyrannosaurus rex* (adult) | AMNH 5027, CM 9380 | *Paul (2010)* and *Hartman's (2013a)* | *Persons & Currie (2011a)* | Photo; *Osborn (1917)* |
| *Tyrannosaurus rex* (adult) | FMNH PR 2081 | *Hartman's (2013a)* | *Persons & Currie (2011a, 2011b)* | Photo; *Brochu (2003)* |
| *Gorgosaurus libratus* (adult) | AMNH 5458, NMC 2120 | *Paul (1988, 2010)* | *Paul (1988)* | Photo; *Paul (2010)* |
| *Gorgosaurus libratus* (juvenile) | AMNH 5664 | *Paul (2010)* | *Paul (1988)* | Photo; *Matthew & Brown (1923)* |
| *Gorgosaurus libratus* (juvenile) | TMP 91.36.500 | *Currie (2003)* and *Hartman (2015)* | *Paul (1988)* | Photo; *Currie (2003)* and *Hartman (2015)* |
| *Daspletosaurus torosus* | CMN 8506 | *Paul (2010)* | *Paul (1988)* and *Russell (1970)* | *Russell (1970)* |

**Notes:**
Institutional abbreviations: AMNH, American Museum of Natural History; BMRP, Burpee Museum (Rockford), Paleontology; CM, Carnegie Museum of Natural History; CMN, Canadian Museum of Nature; CV, Municipal Museum of Chunking; FMNH, Field Museum of Natural History; LH PV, Long Hao Institute of Geology and Paleontology; MOR, Museum of the Rockies; MUCPv, Museo de la Universidad Nacional del Comahue, El Chocón collection; NCSM, North Carolina State Museum; NMC, National Museum of Canada; OUM, Oxford University Museum; PIN, Paleontological Institute, Russian Academy of Sciences; TMP, Royal Tyrrell Museum of Palaeontology; UCMP, University of California Museum of Paleontology; USNM, United States National Museum; UUVP, University of Utah Vertebrate Paleontology; ZDM, Zigong Dinosaur Museum; ZPAL, Paleobiological Institute of the Polish Academy of Sciences.
[†] Different genus used for modified dorsal body outline.

views (Fig. 1). Coordinates were digitized for dorsal, ventral, midsagittal, and lateral contours using PlotDigitizer (*Huwaldt, 2010*), scaled to femur lengths of the specimens. Coordinates were opened as CSV data in Microsoft Excel.

If a dorsal reconstruction of the skeleton was unavailable, a dorsal view of the animal's nearest relative was modified (*Taylor, 2009*). Ideally this relative is the immediate sister taxon or another specimen of the same species but at a different growth stage (as with young *Gorgosaurus* and *Tyrannosaurus*). Anterior and posterior extremes of the head, neck, trunk (coracoids to anterior edge of ilium), ilium, and tail were marked on the lateral view. The corresponding structures on the dorsal view were selected and modified to match their anteroposterior dimensions in the lateral view. Width of the surrogate dorsal view was modified based on literature- or specimen-based width measurements of available structures. For example, many transverse measurements of a juvenile *Tyrannosaurus rex* skeleton (BMR P2002.4.1; courtesy of Scott Williams) were used to modify a dorsal view of an adult (*Persons & Currie, 2011a*). The distal portion of the tail in *Yangchuanosaurus* was modeled on the more complete tail of *S. hepingensis*.

If a dorsal view of only the skull was available for a given dinosaur, and a dorsal view of the skeleton was only available for a related taxon, the differential in skull widths between the taxa was applied to the entire dorsal view of the relative's skeleton. When possible, we used transverse widths of occipital condyles and frontals, measured by author PJC, to confirm ratios of total reconstructed skull widths. The width of the occipital condyle reflects width of the atlas and postaxial cervical vertebrae, and hence influences width of remaining vertebrae as well. This wholesale modification of body width is therefore tentative, but uses the best-constrained available data, and is testable with future, more complete descriptions and measurements of theropod postcrania. We applied this method for dorsal reconstructions of *Sinraptor*, *Eustreptospondylus*, *Dilophosaurus*, *Tarbosaurus*, and one juvenile *Gorgosaurus*. For example, for *Eustreptospondylus* the skull width from *Walker (1964)* was used to modify a dorsal reconstruction of *Allosaurus*, and the skull width of *S. hepingensis* was applied to a dorsal view of its close relative *Y. shangyouensis*. Ribcage width in individual animals varies with ventilatory movements, but width variations of ±10% (*Henderson & Snively, 2003*; *Bates et al., 2009a*) have sufficiently small effect on $I_y$ to permit statistically valid comparisons (see *Henderson & Snively, 2003*).

We also digitized the hind legs of the specimens, by extending their skeletons and soft tissue outlines to obtain anterior and posterior coordinates. We applied a uniform semi-minor axis in the mediolateral direction, as a radius from the midline of the femur to the lateral extent of its reconstructed musculature (*Paul, 1988*, *2010*). The anterior and posterior points on the ilium constrained the maximum anteroposterior extent of the thigh muscles (*Hutchinson et al., 2005*), which we tapered to their insertions at the knee. The anterior point of the cnemial crest constrained the anterior extent of the crural muscles, but the posterior contours were admittedly subjective. In *Paul's (1988*, *2010)* reconstructions, the posterior extent of the m. gasctrocnemius complex in lateral view (bulge of the "drumstick" muscles) generally correlates with the width of the distal portion of the femoral shaft, where two bellies of these muscles originate. Masses of both legs were added to that of the axial body to obtain total body mass. Forelimbs were not included, because they could not be digitized for all specimens and add proportionally little to overall mass moments of inertia (*Henderson & Snively, 2003*; *Bates et al., 2009a*).

The reduced forelimbs of tyrannosaurids would likely add less to overall body $I_y$ than the larger forelimbs of other large theropods, especially with shorter glenoacetabular distance in tyrannosaurids (*Paul, 1988*). However, even the robust forelimbs of *Acrocanthosaurus*, for example, would contribute only 0.15% of the $I_y$ of its entire axial body (*Bates et al., 2009a*).

## Mass property estimates

### Volume and mass

Body volume, mass, COM, and mass moment of inertia were calculated using methods similar to those of *Henderson (1999)*, *Motani (2001)*, *Henderson & Snively (2003)*, *Durkin & Dowling (2006)*, and *Arbour (2009)*. Body segments were approximated as frusta (truncated cones), and volume of the axial body calculated as the sum of volumes of constituent frusta (mass estimates incorporated regional densities of the body; see below). Coordinates for midsagittal and coronal outlines were used to calculate radii for anterior and posterior areas of each frustum. *Arbour (2009)* thoroughly explains the equations and procedures for calculating volume of conical frusta. Equation (1) is for volume of an elliptical frustum, in notation of radii ($r$) and length ($l$).

$$V = \frac{\pi}{3} \times l\left(r_{\text{ant}}^{\text{DV}} r_{\text{ant}}^{\text{LM}} + r_{\text{post}}^{\text{DV}} r_{\text{post}}^{\text{LM}} + \sqrt{r_{\text{ant}}^{\text{DV}} r_{\text{ant}}^{\text{LM}} r_{\text{post}}^{\text{DV}} r_{\text{post}}^{\text{LM}}}\right) \tag{1}$$

The superscript DV refers to a dorsoventral radius, and LM the lateral-to-midsagittal dimension (Fig. 2).

This equation can be generalized to frustum face areas of any cross-section (Eq. (2); similar to equations presented by *Motani (2001)* and *Arbour (2009)*).

$$V = 1/3 \times l\left(\text{Area}_{\text{anterior}} + \text{Area}_{\text{posterior}} + \sqrt{\text{Area}_{\text{anterior}}\text{Area}_{\text{posterior}}}\right) \tag{2}$$

Using Eq. (2), frustum volumes can be calculated from cross-sections departing from that of an ellipse. Vertebrate bodies deviate from purely elliptical transverse sections (*Motani, 2001*). We therefore calculated areas based on a range of superellipse exponents, from 2 (that of an ellipse) to 3 (as seen in whales and dolphins), based on the derivations and correction factors of *Snively (2012)* and *Snively et al. (2013)*. Exponents for terrestrial vertebrates range from 2 to 2.5, with 2.5 being common (*Motani, 2001*; *Snively & Russell, 2007b* used 2.3). *Snively (2012)* and *Snively et al. (2013)* derived and mathematically validated constants for other superelliptical cross-sections; for example, for $k = (2, 2.3, 2.4, 2.5)$, $C = (0.7854, 0.8227, 0.8324, 0.8408)$. Volumes for different cross-sections were then calculated by applying these constants, as superellipse correction factors (*Snively et al., 2013*), to Eqs. (1) and (2).

Frustum volumes were multiplied by densities to obtain masses, and these were summed to obtain axial-body and leg masses. For the head we applied average density of 990 kg/m$^3$, based on an exacting reconstruction of bone and air spaces in *Allosaurus* by *Snively et al. (2013)*. We used a neck density of 930 kg/m$^3$ and trunk density of 740 kg/m$^3$ similar to that of *Bates et al. (2009a)* for the same specimen of *Allosaurus*, which also accounted for air spaces. The post-thoracic and leg densities were set to that of
muscle at 1,060 kg/m$^3$. Density and resulting mass of these anatomical regions was probably greater (even if fat is included) because bone is denser than muscle, which would result in a more posterior COM than calculated here. Rather than introduce new sets of assumptions, we provisionally chose muscle density because its value is known, and the legs (*Hutchinson et al., 2011*) and tail (*Mallison, Pittman & Schwarz, 2015*) have far greater volumes of muscle than bone. All of these density values are easily modifiable in the future, as refined anatomical data for air spaces, bone densities, and bone volumes become available, such as occurred with the restoration methods of *Witmer & Ridgely (2008)* and *Snively et al. (2013)*.

We also varied tail cross-sections by applying the results of *Mallison, Pittman & Schwarz (2015)* for the CFL and full-tail cross-sections of adult *Alligator mississippiensis* and other crocodilians. *Mallison, Pittman & Schwarz (2015)* found that proximal cross-sections of an adult *Alligator* tail and CFL are 1.4 times greater than those previously estimated for young *Alligator* and dinosaurs (*Persons & Currie, 2011a*). We therefore multiplied the original width of the modeled tails of theropods (see above) by 1.4 to obtain an upper estimate of tail thickness and mass.

### Inter-experimenter variation in reconstruction

We checked our mass estimation method against that of *Bates et al. (2009a)* by digitizing their illustrations of *Acrocanthosaurus atokensis*, including the body and the animal's dorsal fin separately. The dorsal fin was restored with half a centimeter of tissue on either side the neural spines, with a bony width of approximately four cm that *Harris (1998)* reported for the twelfth dorsal vertebra. We assumed a rectangular cross-section for the fin. The digitization and mass property estimates (see below) for *Acrocanthosaurus* were purposely carried out blind to the results of *Bates et al. (2009a)*, to avoid bias in scaling and digitizing the outline of their illustrations.

Authors DMH and ES independently digitized reconstructions and estimated mass properties of several specimens, including the legs of many specimens and axial bodies of *Ceratosaurus*, *Allosaurus*, adult *Gorgosaurus*, and *Daspletosaurus*. The software and coding differed in these attempts, and volume reconstruction equations differed slightly (*Henderson, 1999*; *Snively, 2012*; current paper). To further evaluate inter-experimeter variation in results, ES and graduate student Andre Rowe separately used the current paper's methods to digitize an adult *Gorgosaurus* (In all cases discrepancies were negligible, and we were confident to proceed; see Results).

### Centers of mass

To test Hypothesis 1, we calculated anteroposterior and vertical position of the COM of the axial bodies (not including the legs), assuming that the animal would pivot the body around this location if both legs were planted on the ground. First, we calculated the COM of each frustum. Equation (3) gives the anteroposterior position of each frustum's COM (COM$_{AP}$); $r$ are radii of anterior and posterior frusta, and $L$ is its length (usually designated "$h$" for height of a vertical frustum).

Snively et al.
2019
10.7717/peerj.6432

$$\text{COM}_{\text{frustum AP}} = \frac{L \times \left(r_{\text{ant}}^2 + 2r_{\text{ant}}r_{\text{post}} + 3r_{\text{post}}^2\right)}{4 \times \left(r_{\text{ant}}^2 + r_{\text{ant}}r_{\text{post}} + r_{\text{post}}^2\right)} \tag{3}$$

Equation (4) below is an approximation of the dorsoventral position of a frustum's COM ($\text{COM}_{\text{frustum DV}}$), from digitized $y$ (height) coordinates of the lateral body outlines. In this equation, $h_{\text{ant}}$ and $h_{\text{post}}$ are the full heights (dorsoventral dimensions) of the anterior and posterior faces of the frustum, equal to twice the radii $r$ in Eq. (3). The absolute value terms (first and third in the numerator) ensure that the result is independent of whether or not the anterior or posterior face is taller.

$$\text{COM}_{\text{frustum DV}} = \frac{2 \times h_{\text{ant}}\left|h_{\text{post}} - h_{\text{ant}}\right| + h_{\text{ant}}^2 + h_{\text{post}}\left|h_{\text{post}} - h_{\text{ant}}\right| + h_{\text{ant}}h_{\text{post}} + h_{\text{post}}^2}{3 \times h_{\text{ant}} + h_{\text{post}}} \tag{4}$$

Equation (4) gives an exact $\text{COM}_{\text{frustum DV}}$, but assumes that all frustum bases are at the same height (as though they are all resting on the same surface). To obtain the $y$ (vertical) coordinate for the COM of each animal's body, we first approximated $\text{COM}_{\text{frustum DV}}$ using dorsal and ventral coordinates of the anterior and posterior face of each frustum (Eq. (5)).

$$\text{COM}_{\text{frustum DV}} = \frac{\left[\left(y_{\text{ant:dorsal}} + y_{\text{ant:venral}}\right) + \left(y_{\text{post:dorsal}} + y_{\text{post:venral}}\right)\right]}{4} \tag{5}$$

We obtained the COM $\text{COM}_{\text{body}}$ for the entire axial body (both anteroposterior and dorsoventral), by multiplying the mass of each frustum $i$ by its position, summing these quantities for all frusta, and dividing by the entire axial body mass (Eq. (6)). This gives the anteroposterior $\text{COM}_{\text{AP}}$ from the tip of the animal's rostrum, and the dorsoventral $\text{COM}_{\text{DV}}$ at the depth of $\text{COM}_{\text{AP}}$ above the ventral-most point on the animal's trunk (typically the pubic foot).

$$\text{COM}_{\text{body}} = \frac{\sum_{i=1}^{n} \text{COM}_{\text{frustum } i} \times m_{\text{frustum } i}}{m_{\text{body}}} \tag{6}$$

To test Hypothesis 2, we found the position of collective COM of the body and leg, $\text{COM}_{\text{body+leg}}$, which lies lateral to $\text{COM}_{\text{body}}$ calculated in Eq. (6). The lateral ($z$) coordinate of $\text{COM}_{\text{body-}z}$ was set to 0, and that of the leg $\text{COM}_{\text{leg-}z}$ was measured as the distance from $\text{COM}_{\text{body:}z}$ to the centroid of the most dorsal frustum of the leg. Equation (7) enables calculation of $\text{COM}_{\text{body+leg:}z}$ with this distance $\text{COM}_{\text{leg:}z}$, $\text{COM}_{\text{body:}z}$, and the masses of the swing leg and axial body.

$$\text{COM}_{\text{body+leg:}z} = \frac{\text{COM}_{\text{body:}z}m_{\text{body}} + \text{COM}_{\text{leg:}z}m_{\text{leg}}}{m_{\text{body+leg}}} \tag{7}$$

**Mass moments of inertia: Hypothesis 1 (both legs planted)**

Mass moment of inertia for turning laterally, designated $I_y$, was calculated about the axial body's COM by summing individual $I_y$ for all frusta (Eq. (8), first term), and the

contribution of each frustum to the total using the parallel axis theorem (Eq. (8), second term).

$$I_y = \sum_{i=1}^{n} \left(\frac{\pi}{4}\right) \rho_i l_i \bar{r}_{DV} \bar{r}_{LM}^3 + m_i r_i^2 \tag{8}$$

For calculating $I_y$ of an individual frustum, $\rho_i$ is its density, and $l_i$ is its anteroposterior length. The element $\pi/4$ is a constant ($C$) for an ellipse, with an exponent $k$ of 2 for its equation. We modified $C$ with superellipse correction factors for other shapes (*Snively et al., 2013*). The dimension $\bar{r}_{DV}$ is the average of dorsoventral radii of the anterior and posterior faces of each frustum, and $\bar{r}_{LM}$ are the average of mediolateral radii. The mass $m_i$ and COM of each frustum were calculated using the methods described above, and distance $r_i$ from the whole body's COM to that of each frustum was estimated by adding distances between each individual frustum's COM to that of frustum $i$.

### Mass moments of inertia: Hypothesis 2 (pivoting about the stance leg)

Here the body and leg are pivoting in yaw about a vertical axis passing through their collective COM $COM_{body+leg}$, and the center of pressure of the stance foot. Here rotational inertia $I_{y\ body+leg}$ about the stance leg is the sum of the four right terms in Eq. (9).

$$I_{y\ body+leg} = I_{y\ body} + I_{y\ leg} + m_{body} r_{COM\text{-}to\text{-}body}^2 + m_{leg} r_{COM\text{-}to\text{-}leg}^2 \tag{9}$$

Term 1. $I_{y\ body}$ of the axial body about its own COM;

Term 2. $I_{y\ leg}$ of the swing leg about its own COM (assuming the leg is straight);

Term 3. The axial body's mass $m_{body}$ multiplied by the square of the distance $r_{COM\text{-}to\text{-}body}$ from its COM to the collective COM of the body + swing leg ($COM_{body+leg}$);

Term 4. The swing leg's mass $m_{leg}$ multiplied by the square of the distance $r_{COM\text{-}to\text{-}leg}$ from its COM to the collective COM of the body + swing leg ($COM_{body+leg}$).

We calculated $I_{y\ body}$ using Eq. (8). To calculate $I_{y\ leg}$ (Eq. (10)), we approximate the swing leg as extended relatively straight and rotating on its own about an axis through the centers of its constituent frusta. In Eq. (10), $I_{y\ leg}$ is the sum of $I_{y\ frustum}$ for all individual frusta of the leg, and $I_{y\ frustum}$ is in turn simply the sum of $I_x$ and $I_z$ of each frustum (*Durkin, 2003*). These are similar to the first term in Eq. (8), but with anteroposterior radii $r_{AP}$ instead of the dorsoventral radius of frusta of the axial body.

$$I_{y\ leg} = \sum_{i=1}^{n} \left(\frac{\pi}{4}\right) \rho_i l_i (\bar{r}_{AP} \bar{r}_{LM}^3 + \bar{r}_{LM} \bar{r}_{AP}^3) \tag{10}$$

Equations (11) and (12) give distance $r_{COM\text{-}to\text{-}body}$ and $r_{COM\text{-}to\text{-}leg}$ necessary for Eq. (9); note the brackets designating absolute values, necessary to find a distance rather than a $z$ coordinate.

$$r_{COM-to-body} = \left| COM_{body+leg} - COM_{body} \right| \tag{11}$$

$$r_{COM-to-leg} = \left| COM_{body+leg} - COM_{leg} \right| \tag{12}$$
An Excel spreadsheet in Supplementary Information (theropod_RI_body+one_leg.xlsx) has all variables and equations for finding RI of the body plus leg.

## Estimating areas of muscle origination and cross-section

We obtained proxies for muscle force by estimating areas of muscle attachment and cross-section (Fig. 2). Muscle cross-section, and therefore force, scales at a gross level with attachment area for homologous muscles between species, for example with the neck muscles of lariform birds (*Snively & Russell, 2007a*). Enthesis (attachment) size for individual muscles does not scale predictably with force within mammalian species of small body size (*Rabey et al., 2014*; *Williams-Tatala et al., 2016*), which necessitates a more general proxy for attachment area and force correlations between taxa, across spans of evolutionary time (*Moen, Morlon & Wiens, 2016*).

In such interspecific comparisons, morphometrics establish correlation between general muscle origin size and locomotor ecomorphologies (*Moen, Irschick & Wiens, 2013*; *Moen, Morlon & Wiens, 2016*; *Tinius et al., 2018*). Leg length and ilium size are associated with both muscle size and jumping performance in frogs, across biogeography, phylogeny, and evolution (*Moen, Irschick & Wiens, 2013*; *Moen, Morlon & Wiens, 2016*). Between species of *Anolis* lizards, the overall size of muscle attachments on the ilium correlates with necessities of force and moments in different ecomorphotypes, including small and large ground dwellers, trunk and branch climbers, and crown giants (*Tinius et al., 2018*).

In theropods, the ilium is the most consistently preserved element that records leg muscle origination, and is usable for estimating overall origin area of knee extensors, hip flexors, and femoral abductors (Table 2). In large theropods, these enthesis regions have similar gross morphology, including striations indicating Sharpey's fiber-rich origins for the divisions of the m. iliotibialis, and smooth surfaces for the m. iliofemoralis.

Because ilium attachment sites are similar in all theropods, as a reasonable first approximation we infer greater forces for muscles originating from ilia with substantially greater attachment areas than smaller ones (e.g., twice as long and tall). Ilia of large theropod species have a preacetabular flange with a ventral projection, which some authors reconstruct as origin for an anterior head of m. iliotibialis. We include this region in area calculations, but the flange is conceivably also or alternatively an origin for m. iliocostalis, which would stabilize the trunk.

We make similar assumptions for interspecies comparisons of the major femoral retractor, the CFL. The depth of the tail ventral to the caudal ribs correlates with the cross-section of the CFL (*Persons & Currie, 2011a*, *2011b*; *Hutchinson et al., 2011*; *Mallison, 2011*; *Mallison, Pittman & Schwarz, 2015*). Although complete tails are rarely preserved (*Hone, 2012*), the depth of the proximal portion of the tail permits a good first estimation of maximum CFL cross-section (*Persons & Currie, 2011a*, *2011b*; *Mallison, Pittman & Schwarz, 2015*).

Another femoral retractor, the m. caudofemoralis brevis (CFB), originates from the brevis fossa of the postacetabular region of the ilium. We chose to omit the area of origin of the CFB from this analysis, because this would require a ventral view of the ilium,
which is rarely figured in the literature and is difficult to photograph on mounted skeletons. A dorsal view might suffice as a proxy for width of the brevis fossa, but the fossa is flanked by curved alae of bone whose width is obscured in dorsal view. The fossa, and presumably the origination attachment for the CFB (*Carrano & Hutchinson, 2002*), is longer in tyrannosaurids than in other theropods because the ilia are longer relative to body length (*Paul, 1988*), but not broader (*Carrano & Hutchinson, 2002*; figures in *Osborn (1917)*, *Gilmore (1920)*, and *Madsen (1976)*).

Ilium area for muscle attachment was determined for all taxa from lateral-view photographs and scientific illustrations (Table 2) scaled to the size of the original specimen (Fig. 2). Because some muscle scars are ambiguous, the entire lateral surface of the ilium dorsal to the supra-acetabular crest was considered as providing potential area for muscle origination. Images were opened in ImageJ (United States National Institutes of Health, Bethesda, MD, USA), scaled in cm to the size of the original specimens, and the bone areas outlined. ImageJ (under "Measure") was used to calculate areas within the outlines in cm$^2$.

Relative cross-sections were reconstructed for the CFL, although the sample size is smaller than for lateral ilium area, and not large enough for comparative regressions. *Allosaurus*, *Yangchuanosaurus*, several tyrannosaurids, and *Ceratosaurus* have sufficiently well-preserved tails. *Allen, Paxton & Hutchinson (2009)* and *Persons & Currie (2011a)* found that a good osteological predictor of CFL cross-sectional area is vertical distance from the distal tip of the caudal ribs to the ventral tip of the haemal spines. The CFL is never constrained in width to the lateral extent of the caudal ribs, as often previously reconstructed (*Persons & Currie, 2011a*). As a baseline estimate (see Discussion for caveats), we assumed the maximum cross-section to be that at the deepest haemal spine, and that the cross-sections were semi-circular (as ES personally observed in dissections by *Persons & Currie, 2011a*) minus cross-sections of the centra. This method unrealistically simplifies the attachments, ignoring that the lateral and vertical limits of CFL origin are set by the intermucular septum on the caudal ribs between CFL and m. ilioichiocaudalis (*Persons & Currie, 2011b*). Also, simply estimating cross-sections as a proxy for force overlooks functionally and ontogenetically important aspects of intramuscular anatomy, such positive allometry of fascicle length evident in the CFL of *Alligator mississippiensis* (*Allen et al., 2010*). However, as with using the area of the ilium as a proxy for muscle cross-section and force, using tail depth ventral to the caudal ribs is based directly on fossil data. Because the articulations between the haemal arch and caudal centra may not be accurate in skeletal mounts, we varied depths by ±10% to assess their effects on CFL cross-section, and on indices of turning performance. As for our tail cross-section and mass estimates, we also applied the same correction factor of 1.4, that *Mallison, Pittman & Schwarz (2015)* determined for adult *Alligator*, to our estimates of m. caudofemoralis cross-sections, to set an upper bound for cross-section and force.

## Estimates and comparisons of relative agility

We developed two indices of relative agility for theropods: Agilityforce based on agility/force correlations in humans (*Peterson, Alvar & Rhea, 2006*; *Thomas, French & Hayes, 2009*;

*Weiss et al., 2010*), and Agility$_{\text{moment}}$ which incorporates moments or torques. In human studies, maximal muscle force relative to body mass correlates inversely with the time athletes take to complete an obstacle course, which involves rapid changes of direction. Because force is a close direct correlate of agility in humans, independent of torque or power, we were confident in applying force to theropod agility. For Agility$_{\text{force}}$ (Eq. (13)), we divided proxies for overall muscle force (area of muscle origin on the ilium, and cross-section estimates for the CFL) by $Iy$, mass moment of inertia about the $y$ axis through the axial body's COM and a measure of the difficulty of turning the body. This is a comparative index of turning ability, rather than a specific biomechanical quantity.

$$\text{Agility}_{\text{force}} = \frac{A_{\text{ilium}}}{I_y} \tag{13}$$

Here $A_{\text{ilium}}$ is the area (cm$^2$) of the ilium in lateral view. To compare this index of turning ability across theropods, we plotted the results for Agility$_{\text{force}}$ against log10 of body mass for tyrannosaurs and non-tyrannosaurs.

To obtain Agility$_{\text{moment}}$, we first assumed that moment arms scale as mass$^{1/3}$ (an inverse operation of *Erickson & Tumanova's (2000)* Developmental Mass Extrapolation). Mass$^{1/3}$ approximates isometric scaling of moment arms relative to linear size of the animals, which *Bates, Benson & Falkingham (2012)* found to be the likely relationship for allosauroids. Applying this relationship to all of the theropods, we calculated an index of comparative moments, $\tau_{\text{relative}}$, using Eq. (14),

$$\tau_{\text{relative}} = \left( m^{1/3}/100 \right) \times \text{Area}_{\text{ilium}} \times 20 \text{ N/cm}^2, \tag{14}$$

where $m$ is body mass in kg, Area$_{\text{ilium}}$ is ilium area in cm$^2$, and 20 N/cm$^2$ is a sub-maximal concentric specific tension (*Snively & Russell, 2007b*). In SI units, $m^{1/3}$ gives unrealistic moment arms on the order of many meters for larger taxa. Dividing by 100 brings relative moment arms into the more intuitive range of fractions of a meter. This is an arbitrary linear adjustment that (1) does not imply that we have arrived at actual moment arms or torques during life, and yet (2) maintains proportions of $\tau_{\text{relative}}$ among the taxa. Agility$_{\text{moment}}$ is $\tau_{\text{relative}}$ divided by $I_y$ (Eq. (15)), which gives an index of angular acceleration.

$$\text{Agility}_{\text{moment}} = \frac{\tau_{\text{relative}}}{I_y} \tag{15}$$

The quantity $\tau_{\text{relative}}$ does not use actual moment arms, and is not intended for finding angular accelerations. However, our index of relative moment arm lengths is anchored in the isometric scaling of moment arms that *Bates, Benson & Falkingham (2012)* found for allosauroids, and will be testable with more exact estimates from modeling studies. A rich literature directly assesses moment arm lengths in dinosaurs and other archosaurs (*Hutchinson et al., 2005*; *Bates & Schachner, 2012*; *Bates, Benson & Falkingham, 2012*; *Maidment et al., 2013*), and such methods will be ideal for future studies that incorporate estimates of moment arms of individual muscles.

## Visualization of agility comparisons

Although log transformation of mass is useful for statistical comparisons, plotting the raw data enables intuitive visual comparisons of tyrannosaur and non-tyrannosaur agility, and immediate visual identification of outliers (*Packard, Boardman & Birchard, 2009*). We plotted raw agility index scores against log10 body mass in JMP (SAS Institute, Cary, NC, USA), which fitted exponential functions of best fit to the data.

## Statistical comparison of group differences: phylogenetic ANCOVA

Phylogenetic ANCOVA (phylANCOVA) enabled us to simultaneously test the influence of phylogeny and ontogeny on agility in monophyletic tyrannosaurs vs. a heterogeneous group of other theropods. The phylANCOVA mathematically addresses phylogenetically distant specimens or size outliers that would require separate, semi-quantitative exploration in a non-phylANCOVA.

## Phylogenetic approach

All phylogenetically-inclusive analyses were conducted using the statistical program R (*R Core Team, 2015*). For our phylogenetic framework, we used a combination of consensus trees: *Carrano, Benson & Sampson (2012)* for the non-tyrannosauroid taxa (their analyses include the tyrannosauroid *Proceratosaurus*), and *Brusatte & Carr (2016)* for Tyrannosaurioidea, which uses *Allosaurus* as an outgroup. Multiple specimens within the same species (for *Tyrannosaurus rex* and *Tarbosaurus bataar*) were treated as hard polytomies (sensu *Purvis & Garland, 1993*; *Ives, Midford & Garland, 2007*). Basic tree manipulation was performed using the {ape} package in R (version 3.5, *Paradis, Claude & Strimmer, 2004*). Branch lengths were calculated by time-calibrating the resultant tree, as follows. First and last occurrences were downloaded from Fossilworks.org (see Supplementary Information File for Fossilworks citations). Specimens within the same species were further adjusted according to their locality-specific intervals. Time calibration followed the equal-rate-sharing method of *Brusatte et al. (2008)*, which avoids zero-length branches by using a two-pass algorithm to build on previously established methods (*Norell, 1992*; *Smith, 1994*; *Ruta, Wagner & Coates, 2006*). This arbitrarily resolved same-taxon polytomies by assigning near-zero-length branches to the base of each species. The near-zero-length branches effectively maintain the hard polytomy while facilitating transformations of the non-ultrametric variance-covariance matrix.

## Determining strength of phylogenetic signal and appropriateness of phylogenetic regression

To determine whether phylogenetic regression was necessary when analyzing theropod agility, we calculated Pagel's $\lambda$ (*Pagel, 1999*) for each trait examined. Phylogenetic signal was estimated using the R package {phytools} (*Revell, 2010*). We found that phylogenetic signal was high for all traits ($\lambda_{\text{agility force}} = 0.89$; $\lambda_{\text{agility moment}} = 0.90$; $\lambda_{\text{mass}} = 0.88$), emphasizing the need for phylogenetically-informed regression and analysis of covariance.

## Phylogenetically informed analyses

A combination of phylogenetically-informed generalized least squares (PGLS) regression and phylANCOVA was used to test for significant deviations from allometric predictions for both agility force and agility moment (*Garland et al., 1993*; *Smaers & Rohlf, 2016*). The PGLS model calculates the slope, intercept, confidence, and prediction intervals following a general linear model, adjusting expected covariance according to phylogenetic signal (in this case, Pagel's λ; *Pagel, 1999*; for a recent discussion of PGLS methodology, see *Symonds & Blomberg, 2014*). PGLS regression was conducted using the R package {caper} (*Orme et al., 2013*), which implements regression analysis as outlined by *Freckleton (2002)*. We then tested for significant departures from allometry using the recently-derived phylANCOVA method of *Smaers & Rohlf (2016)*. In standard ANCOVA methodologies, comparisons are made outside of a least-squares framework (*Garland et al., 1993*; *Garland & Adolph, 1994*; *Smaers & Rohlf, 2016*). As implemented in the R package {evomap} (*Smaers, 2014*), phylANCOVA compares differences in residual variance in conjunction with the phylogenetic regression parameters (*Smaers & Rohlf, 2016*). This enables a direct least-squares test comparing the fit of multiple grades relative to a single grade (*Smaers & Rohlf, 2016*). We assigned three groups using indicator vectors: Tyrannosauridae, putative juveniles within Tyrannosauridae (hereafter "juveniles"), and non-tyrannosaur theropods (hereafter "other theropods"). GLS standard errors were used to directly test for significant differences in intercept and slope between groups, within a generalized ANCOVA framework (*Smaers & Rohlf, 2016*). We tested the following groupings: (1) Among groups (adult Tyrannosauridae vs. juveniles vs. other theropods); (2) juveniles vs. adult Tyrannosauridae; (3) Tyrannosauridae vs. other theropods. For each of these comparisons, the phylANCOVA applied *F*-tests to partitioned group means. This analysis was performed twice: once for Agility$_{force}$ and again for Agility$_{moment}$.

## Standard for rejecting a null hypothesis of equal agilities

Complications of phylogeny, ontogeny, and biomechanics necessitate a high statistical standard for comparing agility results between sample groups. Reconstructing anatomy and function in fossil animals has potential for many biases—including scaling errors, anatomical judgment in reconstructions and digitizing, fossil incompleteness, and variation in muscle anatomy. If one group appeared to have greater agility than the other, we tested the null hypothesis (no difference) with conditional error probabilities $\alpha(p)$ (*Berger & Sellke, 1987*; *Sellke, Bayarri & Berger, 2001*), a Bayesian-derived standard appropriate for clinical trials in medicine. Conditional error probabilities give the likelihood of false discoveries/false positive results (*Colquhoun, 2014*), effectively the likelihood that the null hypothesis is true, regardless of the original distribution of the data. When $p = 0.05$ in idealized comparisons of only two groups, the probability of false discoveries approaches 29% (*Colquhoun, 2014*). We therefore considered ANCOVA group means to be definitively different if $p$ was in the range of 0.001, at which the probability of a false positive is 1.84% (*Colquhoun, 2014*). We calculated conditional

error probabilities α(*p*) using Eq. (16) (modified from *Sellke, Bayarri & Berger (2001)*), which employs the originally calculated *p*-value from the ANCOVA.

$$\alpha(p) = \left(1 + [-ep\ln(p)]^{-1}\right)^{-1} \tag{16}$$

## RESULTS

### Mass properties and comparison with other studies

Masses, COM, and mass moments of inertia are listed in Tables 3 and 4. "Best estimate" masses (Table 3) are reported for a common cross-sectional shape of terrestrial vertebrates (with a superellipse exponent of 2.3). Here we report and compare individual results, and compare between groups below, under the sections "*Regressions of agility indices vs. body mass*" and "*Results of phylogenetic ANCOVA.*" Inter-experimenter error was negligible. For example, leg masses converged to within 1% when reconstructions were identically scaled, and COM for *Daspletosaurus* was within ±0.4 mm.

Volumes and masses show broad agreement between our results and those calculated in other studies, such as by laser scanning of skeletal mounts (*Bates et al., 2009a*, *2009b*; *Hutchinson et al., 2011*) and fitting splines between octagonal hoops or more complex cross-sections. Our estimates of axial body mass (not including the legs) of *Acrocanthosaurus* ranged from 4,416 kg (elliptical cross-sections with *k* = 2) to 4,617 kg (*k* = 2.3 super-ellipse exponent), compared with the 4,485 kg best-estimate result of *Bates et al. (2009a)*. A slender-model body+legs mass estimate of *Tyrannosaurus rex* specimen FMNH PR 2081 yielded 8,302–8,692 kg depending on superellipse cross-section, compared with *Hartman's (2013b)* GDI estimate of 8,400 kg. A 13% broader model (applying the breadth of the mount's ribcage to our entire dorsal view) yielded 9,131 kg, similar to *Hutchinson et al.'s (2011)* estimate of 9,502 kg (their "lean" reconstruction: *Hutchinson et al., 2011*). Our largest model (Fig. 1), with an anatomically plausible 40% broader tail (*Mallison, Pittman & Schwarz, 2015*) and 13% broader ribcage, yielded 9,713 kg. The current study's results for the juvenile *Tyrannosaurus* BMR 2002.4.1 vary between 575 and 654 kg, from −10% to +2.3% of the 639 kg "lean model" estimate of *Hutchinson et al. (2011)*. Volumes for *Tyrannosaurus* and *Giganotosaurus* are lower than those calculated by *Henderson & Snively (2003)* and *Therrien & Henderson (2007)*, because leg width was narrower in the current study. However, the broad-model volume estimate for the large *Tyrannosaurus* converges with the narrow-ribcage model used in *Henderson & Snively's (2003)* sensitivity analysis, suggesting reasonable precision given inevitable errors of reconstruction.

Relative mass moments of inertia for tyrannosaurids and non-tyrannosaurids did not change with the upper-bound correction factor of 1.4 times the tail cross-sectional area (*Mallison, 2011*; *Mallison, Pittman & Schwarz, 2015*) and mass. However, absolute masses of the entire bodies increased by 5–7% in the tyrannosaurids and most allosauroids, and by 17% in *Acrocanthosaurus*. With this adjustment to tail cross-section, our mass estimates for the *Tyrannosaurus* specimens fell within the lower part of the range that *Hutchinson et al. (2011)* calculated for the largest specimen of this taxon. COM shifted posteriorly by 5–15% (greatest for *Allosaurus*), placing them closer to the anteroposterior location of the

**Table 3 Ilium area, mass properties, and relative agility of theropod dinosaurs.**

| | Ilium area | Total mass | | Mass moments of inertia | | | Agility$_{force}$ axial body | Agility$_{moment}$ axial body | Agility$_{force\ body+leg}$ | Agility$_{moment\ body+leg}$ |
|---|---|---|---|---|---|---|---|---|---|---|
| | A (cm²) | kg | log10 | $I_{y\ body}$ (kg·m²) | $I_{y\ leg}$ (kg·m²) | $I_{y\ body+leg}$ (kg·m²) | A/I | $\tau_{relative}$/I | A/I | $\tau_{relative}$/I |
| **Taxon** | | | | | | | | | | |
| *Dilophosaurus wetherelli* | 380.16 | 372.07 | 2.571 | 213 | 0.279 | 218 | 1.78 | 2.57 | 1.75 | 2.51 |
| *Ceratosaurus nasicornis* | 903.83 | 678.26 | 2.831 | 546 | 1.093 | 559 | 1.60 | 2.21891 | 1.57 | 2.61 |
| *Eustreptospondylus oxoniensis* | 280 | 206.26 | 2.314 | 70.45 | 0.098 | 73.26 | 3.97 | 4.70 | 3.82 | 4.52 |
| *Allosaurus fragilis* | 1,131.5 | 1,512.10 | 3.180 | 2,303.25 | 2.405 | 2,344.62 | 0.49 | 1.13 | 0.48 | 1.11 |
| *Allosaurus fragilis* | 1,228.06 | 1,683.33 | 3.226 | 2,036.81 | 2.121 | 2,078.55 | 0.60 | 1.43 | 0.59 | 1.41 |
| *Acrocanthosaurus atokensis* | 2,551.25 | 5,474.1 | 3.738 | 14,979 | 19.718 | 15,377.24 | 0.17 | 0.60 | 0.17 | 0.58 |
| *Giganotosaurus carolinii* | 3,540.64 | 6,907.6 | 3.839 | 35,821 | 23.731 | 26,593.36 | 0.10 | 0.511 | 0.13 | 0.507 |
| *Sinraptor hepingensis* | 1,268.9 | 2,373.5 | 3.430 | 3,530.7 | 4.929 | 3,740.32 | 0.343 | 0.93 | 0.339 | 0.91 |
| *Yangchuanosaurus shangyouensis* | 992.4 | 2,176.4 | 3.173 | 2,836.7 | 3.365 | 1,672.88 | 0.61 | 1.36 | 0.59 | 1.31 |
| *Raptorex kriegsteini* | 179.7 | 47.07 | 1.673 | 4.65 | 0.0205 | 4.68 | 43.96 | 31.74 | 43.60 | 31.49 |
| *Tarbosaurus bataar* (juvenile) | 1,455.2 | 727.45 | 2.861 | 535 | 1.437 | 548 | 2.72 | 2.39 | 2.65 | 4.77 |
| *Tarbosaurus bataar* (adult) | 2,800 | 2,249.1 | 3.352 | 3,069.9 | 5.586 | 3,126.17 | 0.912 | 2.39 | 0.905 | 2.37 |
| *Tarbosaurus bataar* (adult) | 2,977 | 2,816.3 | 3.450 | 4,486 | 10.049 | 4,515.1 | 0.664 | 1.87 | 0.659 | 1.86 |
| *Tyrannosaurus rex* (juvenile) | 1,107.41 | 660.23 | 2.820 | 344.83 | 0.683 | 347 | 3.21 | 5.59 | 3.19 | 5.56 |
| *Tyrannosaurus rex* (adult) | 4,786.49 | 6,986.6 | 3.844 | 18,175 | 34.067 | 18,276.08 | 0.263 | 1.01 | 0.262 | 1.00 |
| *Tyrannosaurus rex* (adult) | 6,661.8 | 9,130.87 | 3.963 | 28,847 | 51.205 | 29,297 | 0.231 | 0.97 | 0.227 | 0.95 |
| *Gorgosaurus libratus* (adult) | 2,358 | 2,427.3 | 3.385 | 3,219 | 9.79 | 3,312 | 0.73 | 1.97 | 0.70 | 1.67 |
| *Gorgosaurus libratus* (juvenile) | 1,040.56 | 687.7 | 2.837 | 402 | 1.087 | 420.14 | 2.59 | 4.56 | 2.48 | 4.37 |
| *Gorgosaurus libratus* (juvenile) | 1,060.93 | 496.1 | 2.70 | 251.95 | 0.660 | 265.29 | 4.21 | 6.67 | 4.00 | 6.33 |
| *Daspletosaurus torosus* | 3,209.77 | 3,084.8 | 3.489 | 5,338 | 9.665 | 5,586 | 0.60 | 1.75 | 0.58 | 1.67 |

**Note:**

Mass properties are "best estimate" values, assuming superellipse body cross-sections with exponent $k$ = 2.3 (compared with $k$ = 2 for an ellipse). This cross-section is common for terrestrial vertebrates, and has 4.7% greater area than an ellipse of the same radii. Differing exponents, specific tension coefficients for absolute muscle force, and relative moment arms (scaled as body mass$^{1/3}$) do not change relative agilities of tyrannosaurids and large non-tyrannosaurids predatory theropods. Agility$_{force}$ is an estimate of relative maneuverability based on a human athletic standard that finds turning ability is highly correlated with leg muscle force/body mass ratio. Agility$_{moment}$ enables comparison of turning ability by incorporating scaled moment arms for estimating relative torques. As a first approximation, Agility$_{moment}$ assumes similar scaling of moment arms across all taxa.

acetabulum. The COM were anteroposteriorly coincident with the acetabulum in the large-tail models of *Acrocanthosaurus* and *Sinraptor*. With or without an expanded tail, the CM for *Acrocanthosaurus* was found to be consistent with results of *Bates, Benson & Falkingham (2012)*, but to lie posterior to the position estimated by *Henderson & Snively (2003)*.

**Table 4 Centers of mass (COM) and rotation axes for large theropod dinosaurs.**

| Taxon | Axial body COM ($z = 0$) | | Swing leg rotation axis | | Axial body + swing leg rotation axis | |
| --- | --- | --- | --- | --- | --- | --- |
| | $x$ | $y$ | $x$ | $z$ | $x$ | $z$ |
| *Dilophosaurus wetherelli* | 2.33 | 0.42 | 2.61 | 0.17 | 2.36 | 0.02 |
| *Ceratosaurus nasicornis* | 2.66 | 0.50 | 3.07 | 0.15 | 2.70 | 0.01 |
| *Eustreptospondylus oxoniensis* | 1.46 | 0.33 | 1.84 | 0.10 | 1.51 | 0.01 |
| *Allosaurus fragilis* | 2.72 | 0.64 | 3.26 | 0.0.24 | 2.77 | 0.02 |
| *Allosaurus jimmadseni* | 2.64 | 0.79 | 3.20 | 0.23 | 2.69 | 0.02 |
| *Acrocanthosaurus atokensis* | 4.34 | 0.91 | 4.69 | 0.46 | 4.36 | 0.03 |
| *Giganotosaurus carolinii* | 4.54 | 1.33 | 5.10 | 0.44 | 4.57 | 0.03 |
| *Sinraptor hepingensis* | 3.12 | 0.86 | 3.57 | 0.15 | 3.16 | 0.01 |
| *Yangchuanosaurus shangyouensis* | 2.40 | 0.72 | 2.99 | 0.23 | 2.45 | 0.02 |
| *Tarbosaurus bataar* (juvenile)/*Raptorex* | 0.87 | 0.22 | 0.05 | 0.0073 | 0.88 | 0.007 |
| *Tarbosaurus bataar* (juvenile) | 1.93 | 0.54 | 2.33 | 0.15 | 1.98 | 0.02 |
| *Tarbosaurus bataar* (ZPAL) | 2.85 | 0.80 | 0.31 | 0.027 | 2.87 | 0.014 |
| *Tarbosaurus bataar* (adult) | 3.01 | 0.87 | 0.29 | 0.028 | 2.07 | 0.025 |
| *Tyrannosaurus rex* (juvenile) | 2.19 | 0.60 | 0.16 | 0.018 | 2.19 | 0.02 |
| *Tyrannosaurus rex* (adult) | 3.82 | 1.15 | 0.36 | 0.032 | 3.87 | 0.04 |
| *Tyrannosaurus rex* (adult) | 3.84 | 1.17 | 0.40 | 0.040 | 3.90 | 0.04 |
| *Gorgosaurus libratus* (adult) | 3.20 | 0.89 | 3.72 | 0.29 | 3.27 | 0.04 |
| *Gorgosaurus libratus* (AMNH juvenile) | 1.73 | 0.49 | 2.21 | 0.18 | 1.79 | 0.02 |
| *Gorgosaurus libratus* (TMP juvenile) | 2.03 | 0.52 | 2.51 | 0.13 | 2.10 | 0.02 |
| *Daspletosaurus torosus* | 3.35 | 1.16 | 3.93 | 0.25 | 3.43 | 0.05 |

Note:
Axial body: The $x$ value is the position (m) from the anterior tip of the rostrum (where $x = 0$), and $y$ value is the distance (m) from the ventral point of the body ($y = 0$). The $z$ position is 0, at the midline of the body, because the body is modeled as symmetrical. Swing leg: This is the positive $z$ coordinate position (in m) of the leg relative to that of the axial body's COM. Axial body+swing leg: The $z$ coordinate positon (m) of the collective COM of the body and swing leg. The value is small because the leg's mass is much less than that of the axial body.

The largest specimens, *Giganotosaurus carolinii* and the large *Tyrannosaurus rex*, are nearly two tones more massive than their nearest relatives in the sample. The adult *Tyrannosaurus rex* specimens are more massive than *G. carolinii*, corroborating predictions of *Mazzetta, Christiansen & Fariña (2004)* and calculations of *Hartman's (2013b)* for the specimens. The axial body of the reconstructed *Giganotosaurus* specimen is longer, but the large legs and wide axial body of the *Tyrannosaurus rex* specimens contribute to a greater mass overall.

Changing the depth of the tails by ±10% changed the mass of the tails by the same amount, but changed the overall body masses by no more than 3% (less in the tyrannosaurids, which had more massive legs). Varying tail depth changed mass moments of inertia $I_y$ by less than 4%, too small to have an effect on trends in relative $I_y$ in tyrannosaurids vs. non-tyrannosaurids.

Mass moments of inertia including a swing leg were between 0.55% and 5.3% greater than $I_y$ of the axial bodies alone, and agilities correspondingly lower. $I_y$ with the swing leg increased the least with *Acrocanthosaurus*, *Giganotosaurus*, large specimens of *Tarbosaurus* and especially *Tyrannosaurus*, and (surprisingly) *Raptorex*. *Gorgosaurus*

**Table 5 Variation of mass properties with different tail widths.**

| Taxon | Specimen | mass: initial (kg) | mass: 1.4 tail (kg) | CM initial (m from rostrum) | CM (1.4 tail) | $I_y$ (initial) | $I_y$ (1.4 tail) | mass: % initial | CM: % initial | $I_y$: % initial |
|---|---|---|---|---|---|---|---|---|---|---|
| *Tarbosaurus bataar* | ZPAL MgD-I/4 | 2,249 | 2,367 | 2.68 | 2.97 | 3,070 | 3,578 | 105.2 | 110.8 | 116.5 |
| *Tyrannosaurus rex* | AMNH 5027 | 6,986 | 7,458 | 3.82 | 4.01 | 18,175 | 21,395 | 106.7 | 105 | 117.7 |
| *Tyrannosaurus rex* | FMNH PR 2081 | 9,131 | 9,657 | 3.79 | 4.24 | 28,847 | 34,742 | 105.1 | 111.9 | 120.4 |
| *Acrocanthosaurus atokensis* | NCSM 14345 | 5,603 | 6,560 | 4.09 | 4.49 | 14,978 | 22,083 | 117.1 | 109.8 | 147.4 |
| *Allosaurus fragilis* | USNM 4734 | 1,356 | 1,456 | 2.42 | 2.78 | 1,662 | 1,982 | 107.4 | 114.9 | 119.3 |
| *Yanchuanosaurus shangyouensis* | CV 00215 | 1,362 | 1,441 | 2.64 | 2.95 | 1,613 | 1,905 | 105.8 | 111.7 | 118.1 |
| *Sinraptor hepingensis* | ZDM 0024 | 2,428 | 2,588 | 3.12 | 3.37 | 3,694 | 4,374 | 106.6 | 108 | 118.4 |

**Note:**
The last three columns are percentages relative to the baseline values.

juveniles, with proportionally long legs, showed the greatest increase in $I_y$ and drops in agility scores when pivoting on one foot.

## Muscle attachments and cross-sectional estimates

Table 3 reports ilium areas of all specimens, and Table 5 gives tail dimensions and calculated cross-sectional areas for the CFL. Tyrannosaurids have 1.2–2 times the ilium area of other large theropods of similar mass (Table 3); these ratios increase substantially when only axial body mass (total minus leg mass) is considered, because tyrannosaurids have longer and more massive legs.

M. caudofemoralis longus cross-sections vary less than ilium area between the theropods (Table 5). They were slightly greater relative to body mass in most tyrannosaurids, which have deeper caudal centra compared with other theropods. For example, the CFL area of the adult *Tyrannosaurus* specimens had 1.26–1.34 times the cross-sectional areas of the *Acrocanthosaurus* and *Giganotosaurus* specimens of similar respective mass. Increasing the transverse dimensions of the CFL by 1.4 times, after *Mallison, Pittman & Schwarz (2015)*, increases cross-sectional areas by the same factor of 1.4 because tail depth did not change. Increasing tail depth by 10% predictably increased CFL area by 21%, and decreasing tail depth by 10% decreased CFL area by 19%.

## Regressions of agility indices vs. body mass

Figures 3–6 show regressions for the taxa included in Tables 1 and 2. Agility index values for tyrannosaurids are higher than for non-tyrannosaurids of similar body mass. Large tyrannosaurids (between 2 and 10 tones) have at least twice the Agility$_{force}$ or Agility$_{moment}$ values of the non-tyrannosaurids. For theropods in the 300–700 kg range, this gap increases to two to three times greater agility in juvenile tyrannosaurids than in allosauroid adults of similar mass. Comparing specimens of different body masses, tyrannosaurids have similar agility values to those of other theropods about half their size.

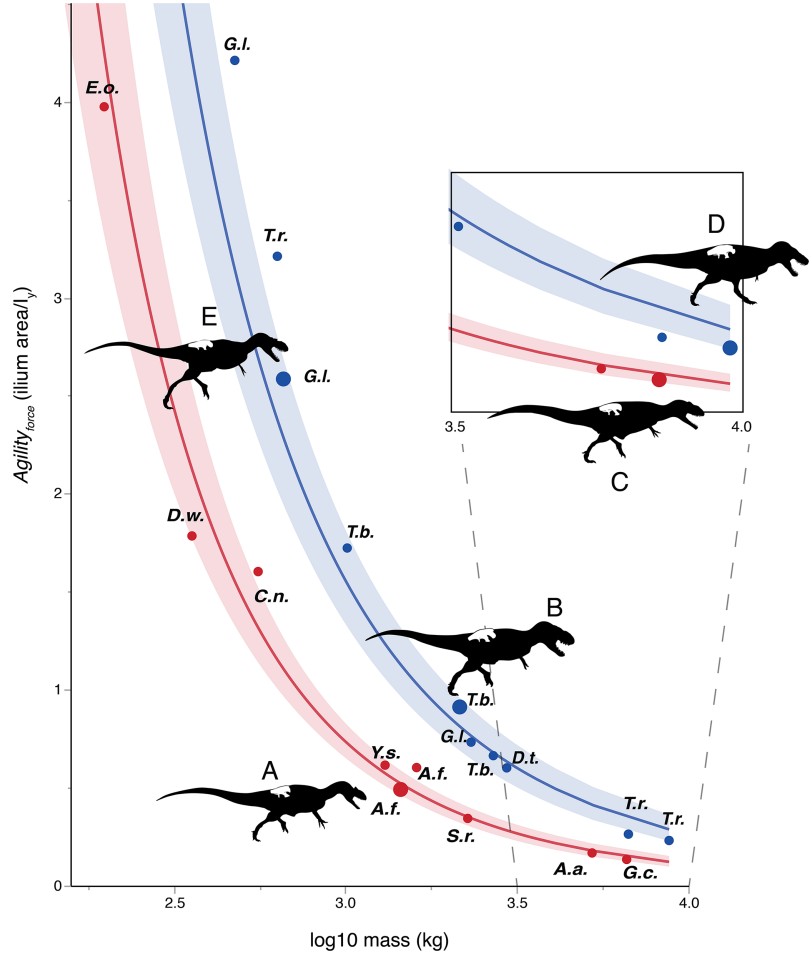

**Figure 3 Log-linear plot of body mass (*x*-axis) vs. an agility index (*y*-axis) based on muscles originating from the ilium, with tyrannosauruids in blue and non-tyrannosaurids in red.** 95% confidence intervals do not overlap. Larger circles show positions of depicted specimens. (A) *Allosaurus fragilis*. (B) *Tarbosaurus bataar*. (C) *Giganotosaurus carolinii* (a shorter-headed reconstruction was used for regressions). (D) *Tyrannosaurus rex*. (E) *Gorgosaurus libratus* (juvenile). The *Tyrannosaurus rex* silhouette is modified after *Hartman (2011)*; others are modified after *Paul (1988, 2010)*. The inset enlarges results for theropods larger than three tones in mass. Note that the tyrannosaurids have two to five times the agility index magnitudes of other theropods of similar mass. Discrepancies between tyrannosaurids and non-tyrannosaurids are greater at smaller body sizes. Abbreviations: *A.a.*, *Acrocanthosaurus*; *A.f.*, *Allosaurus*; *C.n.*, *Ceratosaurus*; *D.t.*, *Daspletosaurus*; *D.w.*, *Dilophosaurus*; *E.o.*, *Eustreptospondylus oxoniensis*; *G.c.*, *Giganotosaurus; G.l.*, *Gorgosaurus*; *S.h.*, *Sinraptor*; *T.b.*, *Tarbosaurus*; *T.r.*, *Tyrannosaurus*; *Y.s.*, *Yangchuanosaurus*.

## Results of phylogenetic ANCOVA

Across all variables, we estimated that much of theropod agility covariance structure can be attributed to phylogenetic affiliation (all $\lambda > 0.88$). The PGLS regression models indicate a strong relationship between agility and mass (Figs. 4 and 5), as well as low variance within agility force ($R^2_{\text{planted}} = 0.9724$; $R^2_{\text{pointe}} = 0.9703$) and agility moment ($R^2_{\text{planted}} = 0.9387$; $R^2_{\text{pointe}} = 0.9384$). The $\lambda$-adjusted PGLS regression line under-predicts agility, fitting non-tyrannosaur theropods more closely than tyrannosaurids (Figs. 4 and 5), indicating that theropods as a whole are more agile than predicted by phylogeny.

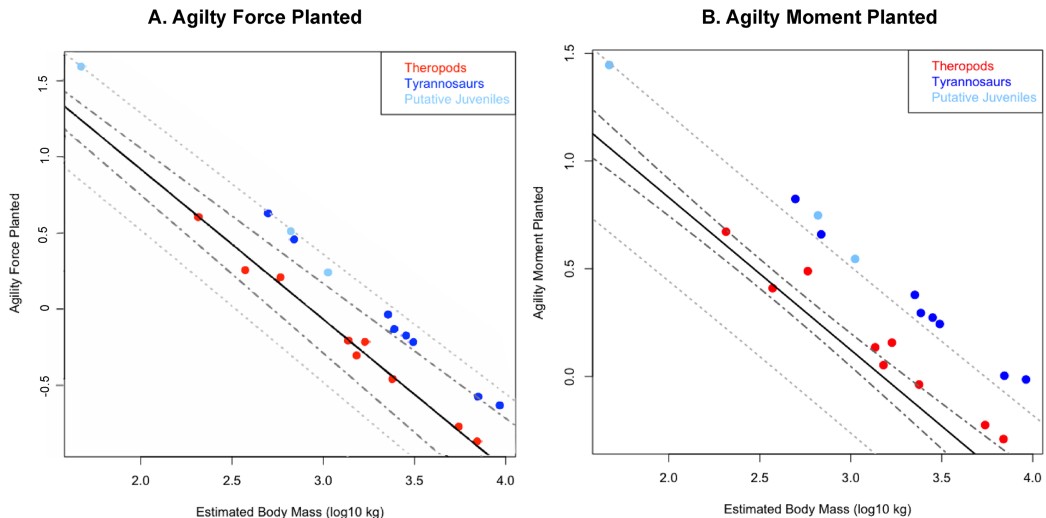

**Figure 4 Phylogenetically generalized least squares regressions of (A) Agility_force and (B) Agility_moment for non-tyrannosaurid theropods (red), adult tyrannosaurids (dark blue), and putative juvenile tyrannosaurids (light blue), turning the body with both legs planted.** Tyrannosaurids lie above or on the upper 95% confidence limit of the regression, indicating definitively greater agility than expected for theropods overall when pivoting the body alone. See Figure, and Supplementary Information Figure and R script, for data point labels.

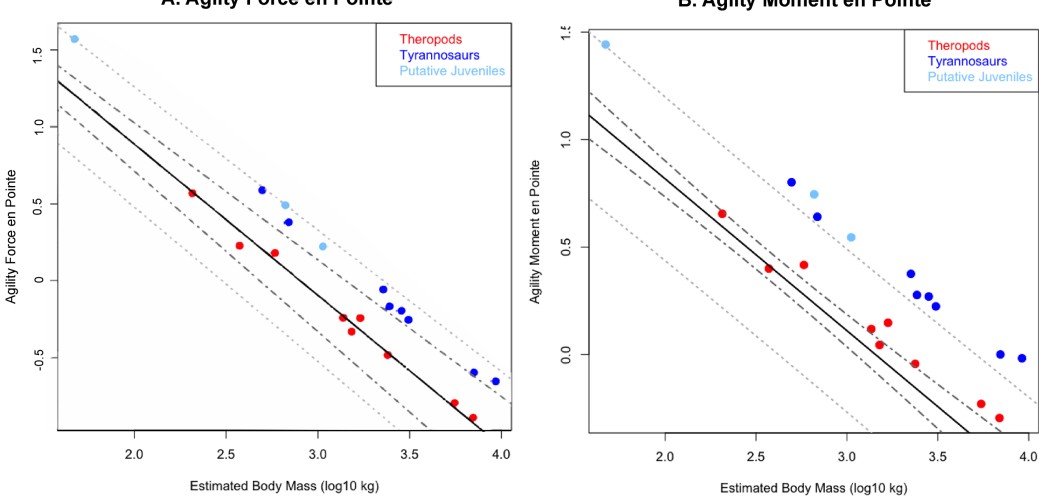

**Figure 5 Phylogenetically generalized least squares regression of (A) Agility_force and (B) Agility_moment for non-tyrannosaurid theropods (red), adult tyrannosaurids (dark blue), and putative juvenile tyrannosaurids (light blue), when pivoting on one leg (en pointe).** Tyrannosaurids lie above or on the upper 95% confidence limit of the regression, indicating definitively greater agility than expected for theropods when pursuing prey. See Fig. 1, and the Supplementary Information Figure and R script, for data point labels.

When 95% confidence and prediction intervals (CI and PI) are calculated according to the phylogenetic variance structure, all tyrannosaurids at or above the 95% PI for all phylogenetic regressions (Figs. 4 and 5).

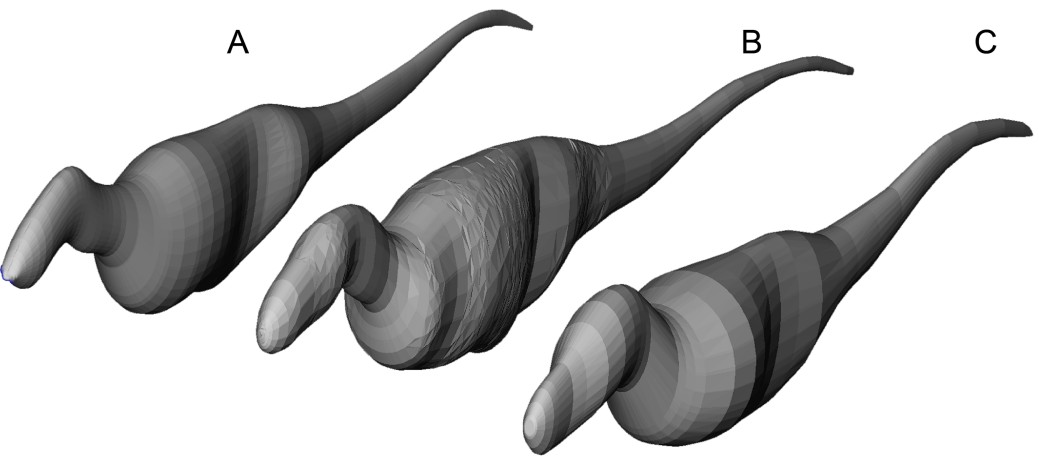

**Figure 6 Axial body models (constructed in FreeCAD) of (A) _Yangchuanosaurus shangyouensis_ (CV 00215), (B) _Sinraptor hepingensis_ (ZDM 0024), and (C) _Tarbosaurus bataar_ (ZPAL MgD-I/4) are within 0.5% of the volumes calculated by summing frusta volumes from Eq. (2).** Three workers built different respective models, and congruence of results suggests low operator variation and high precision between the methods. The _Tarbosaurus_ is lofted from fewer elliptical cross-sections than the others, giving it a smoother appearance that nevertheless converges on the frustum results from many more cross-sections. Note that this is an exercise in cross-validation of volume estimates using uniform densities. Our mass property comparisons use frustum-based calculations that incorporate different densities for different regions of the body.

**Table 6 Regression statistics and comparisons of Agility$_{force}$ and Agility$_{moment}$ between groups of theropods turning their bodies, with both legs planted on the ground.**

**Agility planted**

| Regression statistics | Agility force | Agility moment | Agility force ANCOVA | F | p | a(p) | Agility moment ANCOVA | F | p | a(p) |
|---|---|---|---|---|---|---|---|---|---|---|
| Multiple $R^2$ | 0.861 | 0.939 | Among groups | 10.2 | 0.0014 | 0.0244 | Among groups | 6.71 | 0.077 | 0.3492 |
| Adjusted $R^2$ | 0.853 | 0.935 | Juvenile vs. adult tyrannosaurs | 0.0054 | 0.9421 | 0.1325 | Juvenile vs. adult tyrannosaurs | 0.008 | 0.9301 | 0.1548 |
| Residual standard error | 0.0296 | 0.0197 | All tyrannosaurs vs. theropods | 19.45 | 0.0004 | 0.0084 | All tyrannosaurs vs. theropods | 12.91 | 0.0024 | 0.0379 |
| Degrees of freedom | 18 | 18 | | | | | | | | |
| $F$-statistic | 111.3 | 275.6 | | | | | | | | |
| Slope | −1.01 | −0.708 | | | | | | | | |
| Intercept | 2.92 | 2.246 | | | | | | | | |
| p | $3.9 \times 10^{-9}$ | | | | | | | | | |

**Note:**
Among groups compares all three groups together. Tyrannosaurs vs. juveniles compares adult and juvenile tyrannosaurid specimens, and all tyrannosaurs vs. other theropods combines juvenile and adult tyrannosaurids. Tyrannosaurids have significantly greater agility values than other theropods regardless of grouping, but juvenile and adult tyrannosaurids share an allometric continuum.

Overall, phylANCOVAs for both agility force and agility moment reveal significant differences among all three of our designated groups: tyrannosaurids and putative juveniles vs. other theropods (Tables 6 and 7; $P_{AF\ planted}$ = 0.0014; $P_{AF\ pointe}$ = 0.0013; $P_{AM\ planted}$ = 0.077; $P_{AM\ pointe}$ = 0.0011). When the analysis was broken into specific group-wise comparisons, tyrannosaurids were found to be distinctive from other theropods, whether in the context of agility force or agility moment (Tables 6 and 7;

**Table 7 Regression statistics and comparisons of Agility_force and Agility_moment between groups of theropods turning while pivoting on one foot ("en pointe").**

**Agility en pointe**

| Regression statistics | Agility force | Agility moment | Agility force ANCOVA | F | p | a(p) | Agility moment ANCOVA | F | p | a(p) |
|---|---|---|---|---|---|---|---|---|---|---|
| Multiple $R^2$ | 0.862 | 0.941 | Among groups | 10.44 | 0.0013 | 0.0229 | Among groups | 10.69 | 0.0011 | 0.0200 |
| Adjusted $R^2$ | 0.853 | 0.9373 | Juvenile vs. adult tyrannosaus | 0.149 | 0.7044 | 0.4015 | Juvenile vs. adult tyrannosaus | 0.160 | 0.6938 | 0.4081 |
| Residual standard error | 0.8543 | 0.0192 | All tyrannosaurs vs. theropods | 19.21 | 0.0005 | 0.0102 | All tyrannosaurs vs. theropods | 19.61 | 0.0004 | 0.0084 |
| Degrees of freedom | 18 | 18 | | | | | | | | |
| F-statistic | 112.4 | 284.9 | | | | | | | | |
| Slope | −1.02 | −0.706 | | | | | | | | |
| Intercept | 2.92 | 2.23 | | | | | | | | |
| p | $3.61 \times 10^{-9}$ | $1.73 \times 10^{-12}$ | | | | | | | | |

Note:
Among groups compares all three groups together. Tyrannosaurs vs. juveniles compares adult and juvenile tyrannosaurid specimens, and all tyrannosaurs vs. other theropods combines juvenile and adult tyrannosaurids. Tyrannosaurids have significantly greater agility values than other theropods regardless of grouping, but juvenile and adult tyrannosaurids share an allometric continuum.

$P_{\text{AF planted}} = 0.0004$; $P_{\text{AF pointe}} = 0.0005$; $P_{\text{AM planted}} = 0.0024$; $P_{\text{AM pointe}} = 0.0004$). Putative tyrannosaurid juveniles were not found to be significantly different than their adult counterparts for either performance metric (Tables 6 and 7; $P_{\text{AF planted}} = 0.9421$; $P_{\text{AF pointe}} = 0.7044$; $P_{\text{AM planted}} = 0.9301$; $P_{\text{AM pointe}} = 0.6938$). For this reason, juveniles are not considered apart from adults and have a similar relationship between mass and agility. Conditional error probabilities $\alpha(p)$ for comparisons between groups (Tables 6 and 7) range from 0.008 ($\alpha(p)_{\text{AF planted}}$ and $\alpha(p)_{\text{AM pointe}}$) to 0.038 ($\alpha(p)_{\text{AM planted}}$), indicating low to negligible probabilities of false positive results. Conditional error probability among groups was high for $\alpha(p)_{\text{AM planted}}$, possibly because moment arms scale with mass$^{1/3}$ and have a great influence on the outcome. However, $\alpha(p)$ are still low for both Agility_moment comparison between all tyrannosaurids vs. other theropods.

## DISCUSSION

### Phylogenetic ANCOVA demonstrates definitively greater agility in tyrannosaurids relative to other large theropods examined

Regressions of agility indices against body mass (Figs. 3–5), and especially phylANCOVA (Figs. 4 and 5), corroborate the hypotheses that tyrannosaurids could maneuver more quickly than allosauroids and some other theropods of the same mass.

To evaluate potential biologically-relevant distinctiveness between tyrannosaurids and other theropods, we used a recently developed method of phylANCOVA that enabled group-wise comparisons in the context of the total-group covariance structure (*Smaers & Rohlf, 2016*). By preserving the covariance structure of the entire dataset, this method yields a more appropriate hypothesis test for comparing groups of closely related species (as compared to standard ANCOVA procedures which segregate portions the dataset and therefore compare fundamentally different covariance structures;

*Garland et al., 1993*; *Garland & Adolph, 1994*). Our phylogenetic regression analysis finds that agility and mass are strongly correlated among all theropods ($R^2 > 0.94$; $p < 0.001$), and exhibit a high degree of phylogenetic signal ($\lambda > 0.88$). Using the phylANCOVA of *Smaers & Rohlf (2016)*, we were able to determine that tyrannosaurids exhibit significantly higher agility metrics than other theropods (Figs. 3–5; Tables 6 and 7). Putative tyrannosaurid juveniles were not found to be significantly different from adults and were on or within the 95% prediction interval, aligning these individuals closer to expected phylogenetic structure of their adult counterparts (Figs. 4 and 5; Tables 6 and 7). The slope of the phylogenetic regression lines are greater than −1 but less than 0, suggesting that agility decreases out of proportion to mass as theropods grow, likely because $I_y$ is proportional to the square of body length (*Henderson & Snively, 2003*).

These results allow us to draw important evolutionary conclusions, highlighting the possibility of locomotor niche stratification within Theropoda. The strength of phylogenetic signal combined with the clear degree of separation between tyrannosaurids and non-tyrannosaur theropods underscore the importance of using a phylogenetically-informed ANCOVA to understand between- and within-group agility evolution. By using a phylogenetically-informed analysis, we are able to confirm significant differences in turning behavior, with tyrannosaurs possessing uniquely superior agility scores. These results could indicate a functional specialization for distinctive ecological niches among these groups.

Studies of performance evolution can be difficult because morphology does not always translate into performance differences (*Garland & Losos, 1994*; *Lauder, 1996*; *Lauder & Reilly, 1996*; *Irschick & Garland, 2001*; *Toro, Herrel & Irschick, 2004*). This study, through quantification of multi-body, multifaceted performance metrics, finds strong relationships between morphology, agility, and a distinctive performance capacity by tyrannosaurids. With respect to other theropods, tyrannosaurids are increasingly agile without compromising their large body mass, such that in a pairwise comparison, tyrannosaurids are achieving the same agility performance of much smaller theropods (Figs. 3–5). For example, a 500 kg *Gorgosaurus* has slightly greater agility scores than the 200 kg *Eustreptospondylus*, and an adult *Tarbosaurus* nearly twice the agility scores of the lighter *Sinraptor* This agility performance stratification suggests that these two groups may have had different ecologies, inclusive of both feeding and locomotory strategies. Further, by including juveniles in our analysis through the use of independent inclusion vectors, we were further able to estimate performance capacity in younger life history stages. This revealed that agility performance is established relatively early in life and carries through to large adult body masses.

This quantitative evidence of greater agility in tyrannosaurids is robust, but requires the consideration of several caveats. Agility scores rest on the relationships between agility and muscle force, and muscle force and attachment area. Muscle force and agility correlate directly with each other in humans (*Peterson, Alvar & Rhea, 2006*; *Thomas, French & Hayes, 2009*; *Weiss et al., 2010*), and at a gross level muscle cross-sectional area and force scale with the size of muscle attachments (*Snively & Russell, 2007a*). However, these correlations have yet to be studied in the same system, for example

linking ilium area to force and agility in humans. More thorough testing of the hypothesis will require detailed characterization of muscle sizes, forces and moments in theropods (*Hutchinson, Ng-Thow-Hing & Anderson, 2007*; *Hutchinson et al., 2011*). However, based on dramatic and statistically robust differences between tyrannosaurids and other theropods (Figs. 3–6), we predict that refined studies will corroborate discrepancies in relative agility. Furthermore, we predict that with the same methods, the short-skulled, deep-tailed abelisaurids will have agility indices closer to those of tyrannosaurids than to the representatives of the predominantly allosauroid sample we examined.

**Theropod mass property estimates are consistent between diverse methods, suggesting reliable inferences about relative agility**

Theropod mass and $I_y$ estimates in this study converge with those of other workers, despite differing reconstructions and methods. Our mass estimates for one large *Tyrannosaurus rex* (FMNH PR 2081) are within + or −6% of the "lean" estimate of *Hutchinson et al. (2011)*, who laser scanned the mounted skeleton with millimeter-scale accuracy. *Hutchinson et al.'s (2011)* models of this specimen probably have more accurate dorsoventral tail dimensions than ours, with a relatively greater depth corresponding to that of extant sauroposids (*Allen, Paxton & Hutchinson, 2009*), whereas our models have broader tails. Our mass estimate for the "Jane" specimen (BMR 2002.4.1) was similarly close. These convergences are remarkable, considering that we conducted our estimates long before we were aware of this parallel research, and using a different method. Depending on assumed cross-sections, our axial body estimates for *Acrocanthosaurus* ranged from −1.6% to +2.9% of those of *Bates et al. (2009b)*, which were obtained from laser scanning for linear dimensions, and lofted computer models for volume. As for our estimates of *Tyrannosaurus* mass properties, the *Acrocanthosaurus* calculations were "blind" to *Bates et al.'s (2009a)* results for this specimen. For all of the examined taxa, volumes of the neck and width of the base of the tail are likely greater in our study than in others, even with robust models in their sensitivity analyses (*Hutchinson, Ng-Thow-Hing & Anderson, 2007*; *Bates et al., 2009a*, *2009b*), because our models incorporate new anatomical data on soft tissues (*Snively & Russell, 2007b*; *Allen, Paxton & Hutchinson, 2009*; *Persons & Currie, 2011a*; *Mallison, Pittman & Schwarz, 2015*) indicating a taller, broader neck and broader tail cross-sections. Despite these discrepancies in soft tissue reconstruction, high consistency with methods based on scanning full-sized specimens engenders optimism about the validity of frustum-method estimates (*Henderson, 1999*), despite their dependence on 2D images, restoration accuracy, and researcher judgments about amounts of soft tissue.

Frustum and graphical double integration (GDI) methods also yielded similar results (Appendix 1). When superellipse correction factors were applied to the 9.2 m$^3$ GDI volume *Hartman's (2013b)* obtained for the *Tyrannosaurus rex* (PR 2081), results closer to our broad-bodied volume estimate for the specimen were generated. Assuming a super-ellipse exponent of 2.3, scaling *Hartman's (2013b)* estimate by the correction factor of 1.047 gives an estimate of 9.632 m$^3$, less than 2% greater than our estimate. Furthermore,
applying super-ellipsoid cross-sections may reconcile careful GDI estimates, such as *Taylor's (2009)* for the sauropods *Brachiosaurus* and *Giraffatitan*, with volumes evident from laser scans and photogrammetry of fossil mounts (*Gunga et al., 2008*; *Bates et al., 2016*).

In addition to convergence of mass and volume estimates, different algorithms for COM give nearly identical COM estimates for *Giganotosaurus*, the longest theropod in the sample (see Appendix 1). The discrepancy of only 0.2 mm is negligible for a 13 m-long animal. Although we recommend finding the anteroposterior COM of each frustum using our Eq. (4) (especially for rotational inertia calculations), the simpler approximation method is adequate.

Calculation methods probably have a smaller effect on COM estimates than anatomical assumptions concerning restoration, and variations in the animal's postures in real time. Such postural changes would include turning or retracting the head, and movements of the tail (*Carrier, Walter & Lee, 2001*) using axial (*Persons & Currie, 2011a*, *2011b*, *2012*) and caudofemoral muscles (*Bates et al., 2009b*; *Allen et al., 2010*; *Persons & Currie, 2011a*, *2011b*, *2012*; *Hutchinson et al., 2011*). The congruence of results from different methods is encouraging, because biological factors govern the outcome more than the choice of reconstruction method.

## Relative agilities are insensitive to modeling bias

Reconstruction differences between this and other studies are unlikely to bias the overall comparative results so long as anatomical judgments and methods are consistently applied to all taxa. For example, although tail width is reconstructed similarly in this study and the dissection-based studies of *Allen et al. (2010)* and *Persons & Currie (2011a)*, the tail depths of our models may be too shallow (*Allen et al., 2010*). Consistently deeper tails, better matching reconstructions of *Allen et al. (2010)*, *Bates et al. (2009a*, *2009b)*, and *Hutchinson et al. (2011)*, would, however, not alter our overall comparative results.

Considering $I_y$ and mass from independent studies is instructive in relation to potential modeling bias and error. *Bates et al. (2009b)* calculated notably high mass and $I_y$ (*Hutchinson et al., 2011*) for a *Tyrannosaurus rex* specimen (MOR 555) not included in our study, yet with its enormous ilium its agility indices would be higher than those of a non-tyrannosaurid *Acrocanthosaurus* of equivalent mass (*Bates et al., 2009b*). $I_y$ and agility for the *Allosaurus* examined by *Bates et al. (2009a)* are similar to those for other *Allosaurus* specimens. Consistent modeling bias for all theropods (making them all thinner or more robust) would have no effect on relative agility assessments. Overlap of agility would require inconsistent bias in this study and those of other workers, with more robust tyrannosaurid reconstructions and slender non-tyrannosaurids. This bias is unlikely, because reconstructions were checked against skeletal measurements and modified when necessary, and most reconstructions were drawn from one source (*Paul, 2010*).

Furthermore, the current mass estimates cross-validate those of *Campione et al.'s (2014)* methods based on limb circumference-to-mass scaling in bipeds. Our lower mass estimate (6,976 kg) for one adult *Tyrannosaurus rex* specimen (AMNH 5027) coincides

remarkably with their results (6,688 kg), considering the large tail width of our reconstruction. These close correspondences of inertial properties between different studies gives confidence for biological interpretation.

## Behavioral and ecological implications of agility in large theropods

This discrepancy in agility between tyrannosaurids and other large theropods raises specific implications for prey preference, hunting style, and ecology (*Holtz 2002*; *Holtz, 2004*). By being able to maneuver faster, tyrannosaurids were presumably more adept than earlier large theropods in hunting relatively smaller (*Hone & Rauhut, 2010*), more agile prey, and/or prey more capable of active defense. This capability in tyrannosaurids is consistent with coprolite evidence that indicates tyrannosaurids fed upon juvenile ornithischians (*Chin et al., 1998*; *Varricchio, 2001*), and with healed tyrannosaurid bite marks on adult ceratopsians and hadrosaurs (*Carpenter, 2000*; *Wegweiser, Breithaupt & Chapman, 2004*; *Happ, 2008*). Tyrannosaurids co-existed with herbivorous dinosaurs that were predominately equal to or smaller than them in adult body mass. The largest non-tyrannosaurids, including *Giganotosaurus*, often lived in habitats alongside long-necked sauropod dinosaurs, the largest land animals ever. These associations suggest that allosauroids may have preferred less agile prey than did tyrannosaurids. It is also possible that stability conferred by high rotational inertia, as when holding onto giant prey, was more important for allosauroids than turning quickly.

These faunal correspondences between predator agility and adult prey size are not absolute, however. Tyrannosaurids sometimes shared habitats with large sauropods (Nemegt, Ojo Alamo, and Javalina Formations: *Borsuk-Białynicka, 1977*; *Lehman & Coulson, 2002*; *Sullivan & Lucas, 2006*; *Fowler & Sullivan, 2011*), and even with exceptionally large hadrosaurids (*Hone et al., 2014*). Relative agility of herbivorous dinosaurs must be tested biomechanically to assess the possible advantages of agility in tyrannosaurids. *Snively et al. (2015)* calculated that ceratopsians had lower $I_y$, and hadrosaurs and sauropods greater $I_y$, than contemporaneous theropods, but musculoskeletal turning ability has yet to be assessed in detail for dinosaurian herbivores.

Tyrannosaurids were unusual in being the only toothed theropods (thus excluding large-to-giant oviraptorosaurs and ornithomimosaurs) larger than extant wolves in most of their habitats (*Farlow & Holtz, 2002*; *Farlow & Pianka, 2002*; *Holtz, 2004*). Among toothed theropods, adult tyrannosaurids of the Dinosaur Park Formation were 50–130 times more massive than the next largest taxa (troodontids and dromaeosaurids: *Farlow & Pianka, 2002*). Comparing the dromaeosaur *Dakotaraptor steini* (*DePalma et al., 2015*) and *Tyrannosaurus rex* in the Hell Creek formation reveals an instructive minimum discrepancy. We estimate the mass of *Dakotaraptor* to be 374 kg, using the femoral dimensions provided by *DePalma et al. (2015*: Fig. 9*)* and the equations of *Campione et al. (2014)*. Adult *Tyrannosaurus* attained 17–24 times this mass (our estimates), approximately the difference between a large male lion and an adult black backed jackal. By our estimates, the juvenile *Tyrannosaurus* in our sample was nearly twice as massive as an adult *Dakotarapor*.

These size differences between adult tyrannosaurids and non-tyrannosaurid predators suggest that subadult tyrannosaurids were able to capably hunt midsized prey, in ecological roles vacated by less-agile, earlier adult theropods of similar body mass. In contrast, many earlier faunas (*Foster, Holtz & Chure, 2001*; *Farlow & Holtz, 2002*; *Farlow & Pianka, 2002*; *Russell & Paesler, 2003*; *Holtz, 2004*; *Foster, 2007*; *Läng et al., 2013*; although see *McGowen & Dyke, 2009*) had a continuum of body masses between the largest and smallest adult theropods, and perhaps greater subdivision of niches between adults (*Läng et al., 2013*). Ongoing research (*Shychoski, Snively & Burns, 2011*) evaluates alternative evolutionary scenarios and soft-tissue evidence in a further exploration of tyrannosaurid agility.

# APPENDIX 1

## How precise are different methods of mass property estimation?

In addition to our mathematical slicing procedures (*Henderson, 1999*), methods for calculating mass properties include use of simplified B-splines or convex hulls to represent body regions (*Hutchinson, Ng-Thow-Hing & Anderson, 2007*; *Sellers et al., 2012*; *Brassey & Sellers, 2014*; *Brassey et al., 2016*), or more complex non-uniform rational B-spline reconstruction modified to fit the contours of mounted skeletons and inferred soft tissues (*Bates et al., 2009a*, *2009b*; *Mallison, 2007*, *2010*, *2014*; *Stoinski, Suthau & Gunga, 2011*). *Brassey (2017)* reviews and compares these methods in detail. Both spline-based and mathematical slicing methods have been validated for living terrestrial vertebrates (*Henderson, 1999*, *2004*, *2006*; *Henderson & Snively, 2003*; *Hutchinson, Ng-Thow-Hing & Anderson, 2007*; *Bates et al., 2009a*). However, spline-based methods (as in *Mallison's (2007*, *2010*, *2014)* and similar procedures) are conceivably more accurate than slicing methods, which are based on a few extreme coordinates of the body, and estimate intermediate contours as ellipses or non-ellipsoid superellipses (*Henderson, 1999*; *Motani, 2001*; *Henderson & Snively, 2003*; *Arbour, 2009*; *Snively et al., 2013*). We compared results of mathematical slicing and spline methods by obtaining inertial properties from both slicing abstractions and spline models of several theropods, based on the dimensions used in the slicing calculations.

Another method, termed GDI (*Jerison, 1973*), uses elliptical cylinders instead of frusta to estimate volumes. For reptiles with cylindrical bodies, GDI approximates mass better than regressions based on body length or bone dimensions (*Hurlburt, 1999*). Masses and $I_y$ were calculated by GDI for all specimens, and compared to results from the frustum method.

## Methods for testing precision of mass property results from different approaches

To compare slicing and spline-based inertial property results of full axial bodies of theropods, we constructed spline models of *Y. shangyouensis*, *S. hepingensis*, and *Tarbosaurus bataar* (Fig. 6), after *Snively et al. (2013*, *2015)*. We used FreeCAD (freecadweb.org) to construct the bodies from lofted ellipses, and MeshLab

(meshlab.sourceforge.net) to obtain volume, COM, and the inertia tensor, assuming uniform densities.

We further estimated volumes of *E. oxoniensis* and *Y. shangyouensis* using the GDI methods of *Jerison (1973)*, *Hurlburt (1999)*, *Murray & Vickers-Rich (2004)*, and *Taylor (2009)*, using Eq. (17).

$$V_{\text{body}} = \sum_{n=1}^{i} V_i = \pi(r_{i1})(r_{i2})L_i \qquad (17)$$

The body is divided into segments from 1 to $i$. Each body segment is treated as an elliptical cylinder with the cross-sectional area of its anterior ellipse, with major and minor radii of $r_1$ and $r_2$. This area is multiplied by $L$, the segment's length as the distance to the subsequent ellipse.

We also tested convergence of body COM approximations using COM of each frustum (Eq. (4)), vs. simply assuming that each frustum's anterioposterior COM was very close to its larger-diameter face. The longest specimen, *G. carolinii*, was the best candidate for this test because $I_y$ is sensitive to the square of the distance $r$ (Eq. (8)) of a segment's COM from the body total COM. The distance of the large-diameter face from the animal's rostrum was used as the value for $COM_{\text{frustum}}$ in Eq. (7).

## Results of methods comparison

Values of mass and mass moment of inertia varied little between methods using frusta (truncated cones), extruded ellipses (GDI), and spline (3D lofting) methods. Volumes, COM, and $I_y$ (assuming uniform density) were within 0.5% of each other for frustum and spline models of *S. hepingensis*, *Y. shangyouensis*, and *Tarbosaurus bataar* (Fig. 6). The GDI mass and $I_y$ for *E. oxoniensis* were only 0.1% higher than calculated by the frustum method, and that for *Y. shangyouensis* only 0.5% higher. However, differences increase substantially for estimates of hind limb mass. GDI-calculated mass for the hind leg of *Eustreptospondylus* is over 11% greater than that from the frustum method.

GDI and frustum estimates are closest for axial bodies of the theropods, but diverged for the hind legs. This suggests high accuracy of the method for relatively tubular objects, such as the bodies of some sprawling tetrapods (*Hurlburt, 1999*), and the necks, tails, and legs of giant long-necked sauropod dinosaurs (*Taylor, 2009*). GDI with extruded ellipses is less accurate for highly tapered objects, such as the hind legs of theropods, the trunks of some large theropods and sauropods, and other animals with ribcages that flare laterally in coronal section. However, the high frequency of body cross-sections (*Motani, 2001*), as in our axial body models, ameliorates the potential error of GDI for tapered objects.

For the *Giganotosaurus* model, the position of $COM_{\text{body}}$ from the tip of the rostrum was identical to three significant figures, whether using Eq. (4) or assuming that each frustum's COM was very close to its larger face (4.65665 vs. 4.65685 m, a difference of $2 \times 10^{-4}$ m).

## ACKNOWLEDGEMENTS

We thank Michael Habib and an anonymous reviewer for enlightening critiques, the academic editor for evaluations, plus the Currie, Cotton, Witmer, O'Brien, Gignac, and rex labs for discussion. Scott Williams (Museum of the Rockies) contributed additional measurements. Students Andre Rowe, Erin Wick, Ryan Sokup, Peter Roth, Emily Schneider, and Julianna Cruz (University of Wisconsin-La Crosse) assisted with quantifying variation in the mass property methods.

### Funding

This study was funded by an Alberta Ingenuity Postdoctoral Fellowship (Eric Snively) and Canada Foundation for Innovation grants (Philip J. Currie) at the University of Alberta, National Science Foundation (Lawrence Witmer), University of Wisconsin-La Crosse (Eric Snively), Oklahoma State University (Haley O'Brien), plus Russ College of Engineering, Department of Biological Sciences, and the School of Rehabilitation and Communication Sciences at Ohio University (Eric Snively). The funders had no role in study design, data collection and analysis, decision to publish, or preparation of the manuscript.

### Grant Disclosures

The following grant information was disclosed by the authors:
Alberta Ingenuity Postdoctoral Fellowship.
Canada Foundation for Innovation grants.
University of Alberta, National Science Foundation.
University of Wisconsin-La Crosse.
Oklahoma State University.
Russ College of Engineering, Department of Biological Sciences, and the School of Rehabilitation and Communication Sciences at Ohio University.

### Competing Interests

The authors declare that they have no competing interests.

### Author Contributions

- Eric Snively conceived and designed the experiments, performed the experiments, analyzed the data, contributed reagents/materials/analysis tools, prepared figures and/or tables, approved the final draft.
- Haley O'Brien conceived and designed the experiments, performed the experiments, analyzed the data, contributed reagents/materials/analysis tools, prepared figures and/or tables, approved the final draft.
- Donald M. Henderson conceived and designed the experiments, performed the experiments, analyzed the data, contributed reagents/materials/analysis tools.
- Heinrich Mallison analyzed the data, authored or reviewed drafts of the paper.

- Lara A. Surring authored or reviewed drafts of the paper.
- Michael E. Burns authored or reviewed drafts of the paper.
- Thomas R. Holtz Jr authored or reviewed drafts of the paper.
- Anthony P. Russell authored or reviewed drafts of the paper.
- Lawrence M. Witmer contributed reagents/materials/analysis tools, authored or reviewed drafts of the paper.
- Philip J. Currie contributed reagents/materials/analysis tools, authored or reviewed drafts of the paper.
- Scott A. Hartman prepared figures and/or tables.
- John R. Cotton contributed reagents/materials/analysis tools, authored or reviewed drafts of the paper.

## Data Availability

The raw data is available as Supplemental Files.

## Supplemental Information

Supplemental information for this article can be found online at http://dx.doi.org/10.7717/peerj.6432#supplemental-information.

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
