# Peer review of "Lower rotational inertia and larger leg muscles indicate more rapid turns in tyrannosaurids than in other large theropods"

_PeerJ, doi:10.7717/peerj.6432_

## Round 0.1 · original submission · Major Revisions

The reviewers (and I) are enthusiastic about the innovative approach taken to the topic of agility in tyrannosaur dinosaurs. The phylogenetic accommodations are particularly important, and really help bolster the conclusions of the paper. Our major suggestions center on clarification of terminology as well as clarification on what particular metrics (e.g., Agilityforce) represent in a biomechanical sense.

Overall, the manuscript is on that line between minor and major revisions, but I chose "major revisions" on account of the potentially large impact some of the reviewer suggestions may have. My hope, however, is that they can be easily addressed in revision.

- The issue of ensuring correct terminology in agility vs. maneuverability (especially as noted by reviewer 2), and it should be addressed in revision.

- Although Reviewer 1 suggests trimming the methods section, my personal preference would be to leave it at its current length (after appropriate revision, of course). I feel that it is important for papers to be stand-alone units, and anything to pull all of the relevant information into one place is a good thing. Although it is a major portion of the paper's length, it also ensures maximum transparency.

- Reviewer 1 expressed methodological concerns with how Agilityforce and Agilitymoment were interpreted (i.e., what are they actually testing relative to what they are supposed to test?). The comments from Reviewer 1 on estimating muscle force from insertion area also should be addressed in some detail. Neither the reviewer nor I think the concern sinks the manuscript, but I do think some appropriate explanation and potential reworking of this part of the manuscript is warranted.

OTHER NOTES FROM EDITOR:
- I am not a huge fan of numbered titles (e.g., "Tyrannosaurid agility I"); the number doesn't provide any real information, and is subject to problems if parts II, III, etc., are never published or are published elsewhere, or if they have the titles changed by editors/journals who don't like numbered titles. I have no doubt the authors will follow through on additional contributions for this topic, but nonetheless suggest trimming the title to "Rotational inertia and leg muscles indicate more rapid turns in tyrannosaurids than in other large theropods". My suggestion is totally optional, of course, and I will leave it up to the authors as to a decision.

- For the supplemental Excel file, I am a little uneasy with the instruction to contact an author for instructions on how to use it. Is there anything that can't be included in the manuscript itself? Although I have no doubt that the author will be helpful for anyone who contacts him, this will be problematic at the inevitable point (hopefully many decades distant) when Dr. Snively is no longer available via email. Is there a published paper to refer to?

- Very nice figures!

- In Figure 4, it is noted that some silhouettes are after work by Greg Paul. Please confirm that you have permission to use them here in a CC-BY publication.

- Please include the R files (e.g., CSV, or whatever format) used for the analysis, as supplemental information. It is critical that you also include the tree topology used here (at a minimum as a figure; ideally as both a figure and the appropriate R input file). If you have an R script, or a sequence of commands used to run the analyses, these should also be provided.

Reviewer 1 ·

Basic reporting

In an attempt to stick to the PeerJ reviewer format, I will put my general thoughts and comments on the paper in the final section, but I encourage the authors to read them first, before my more critical comments, which follow below.

My largest comment in terms of basic reporting is that many in-text citations are missing from the bibliography. Here is a list I jotted down, but is likely not comprehensive:

Lu et al 2014
Snively and Russel 2003
Holtz 1995
Peterson et al 2006
Thomas et al 2009
Weiss et al 2010
Snively et al 2013
Mallison et al 2015
Snively 2012
Trinkhaus et al 1991
Allen et al 2010

Of these, three citations are missing from the bibliography that I find a crucial omission, on which the premise of the current study is based. On lines 80-82, it is stated that maneuverability is directly correlated with maximal muscle force in humans. Three studies are cited, none of which are in the bibliography, and briefly searching for them myself I was unable to find them. This is a crucial issue, as this entire study assumes that attachment area is equal to muscle cross sectional area, muscle cross sectional area is proportional to muscle force generation, and muscle force generation is proportional to some metric of agility or maneuverability. I will point out in the experimental design why this is so important. Similarly, the next sentence states that maximal muscle force is straightforward to estimate from fossils as opposed to muscle moment arms, which are difficult. Young et al. 2002 is cited here as evidence of this, but it is missing from the bibliography. Furthermore, I don’t necessarily agree with the statement. I admit that I am not a dinosaur expert, but there are many taxa and/or specific bones for which attachment site morphology is not a good proxy for muscle area or mass (e.g. Williams-Hatala et al., 2016, Scientific Reports; Rabey et al., 2014 Journal of Human Evolution), and similarly muscle moment arms (depending on the muscle) can be estimated from bones alone, and indeed has already been done in some of the taxa investigated here.

One of thing I’d like to point out in terms of basic reporting. The methods and approach used in this paper will likely be of interest to functional morphologists outside of the dinosaur realm, and because of this, I would very strongly encourage the authors to add a figure of the phylogenetic tree of the taxa utilized in this study. Even more helpful would be a phylogenetic tree that also has skeletal outlines that more or less approximate how much material is preserved from each specimen. There is some degree of presumed knowledge of taxa when discussing the methods, and it would be a fairly easy inclusion that would make this paper much easier to read and interpret.

I found that Figure 3 was not a crucially needed figure. In fact, it is only relevant for interpreting the appendix, and I think should not be included as a main figure. At the very least it should not occur before the result figures.

I also found that the methods section was entirely too long (1/2 the total manuscript), and in the experimental design section and general comments section, I have some suggestions for where superfluous information can be reduced.

I have some confusions about the tables and results sections as well. In the results, ranges of body masses are listed that don’t match those in the tables. For instance Arcocanthosaurus was listed as having a range of body mass or 4416-4485 kg, but in table 2, it is listed as 5474.1 and in Table 4 it is listed as 5603. I understand that the authors went through lots of permutations of models, but it needs to be made more explicit which are reported in the tables, and which are reported in the text. The values in the text seem to be lower than those in the table. Furthermore in the table it is unnecessary to report values (especially cm^2) out to 2 significant digits after the decimal. Same with body mass.

I am unsure why the authors chose to use MMI to abbreviate for moment of inertia, considering I is the well accepted abbreviation of this term (and is actually used quite often in the paper).

In the text it is stated that Table 4 gives tail dimensions and cross-sectional areas for the caudofemoralis muscle, but I do not see this in Table 4.

Experimental design

Methods:
The methods section mentions how some outlines were modified based on recent reconstructions, and that for some taxa, there was no dorsal view, so a dorsal view from a close relative was utilized. It would be helpful to know for how many of taxa this was the case. For instance, if only one tyrannosaurid had a dorsal view, and you substituted that dorsal view in for all others, then maybe the signal that was picked up is a methodological issue. Similarly, in the paragraph starting on line 165, it says “if only the skull was available”. I assume this to mean if only a dorsal view of the skull is available, because the specimens that were chosen were supposed to be fairly complete. I think wording here should be changed to make this clearer. But, even so, I would like to know for how many taxa this was an issue, where the entire body plan had to be scaled to the differential between skull measurements. In addition, the following sentence (166-168), implies too much confidence in this method. Not having measurements, and just scaling differences in skull width does not “avoid subjective judgements of widths elsewhere in the body”, it in fact is a process that is a subjective and simplistic judgement of width everywhere else in the body. Furthermore, what does “reconstructs width adequately for MMI comparisons” mean? How do you evaluate this statement, we don’t know what the actual moment of inertia of these animals were, so how can you say that you’ve reconstructed them accurately. This type of language actually pops up quite often (language which implies a degree of confidence in the data that is not there) and I have tried to point some of those instances out.

Reconstructions of the legs:
In the manuscript, it is said that the legs were extended, and the soft tissue outlines were digitized, but how were these soft tissue lines determined or verified? I understand creating a soft tissue outline for the upper body, but for the lower limbs, how was the amount of musculature reconstructed. Furthermore, I understand how an ellipse can then be fitted to the limb in the A/P direction, but how was the minor axis determined in the M/L direction?

I was also initially confused as to the equation for calculating the volume of a frustrum. Eq 1 does not match Fig 1 d, or the equation used by Arbour. I now see that this is a generalized equation that you then simplify to Eq 3, which is utilized. I think the derivation of equation 3 is confusing and unnecessary. Lines 191-204 can essentially be removed without any problem. Just cite Arbour for Eq 3 and move on.

In lines 223-230, it might be worth mentioning that including bone densities would shift the CoM further posteriorly, since it was assumed that the tail and legs were solely muscle density.

The paragraph in lines 238-245: I’m unsure why you have a paragraph dedicated to the comparison of Acrocanthosaurus’ body mass compared to that of Bates, when in the results, you actually compare body masses of several taxa to several other studies. Why is the study of Bates so special as to deserve a whole paragraph dedicated to the fact that you will compare it? However, if a special technique was needed to model the fin then you should put that in the preceding paragraph.

Calculation of center of mass:
In the preceding sections, it was mentioned that the volumes of the legs were calculated in an extended position. In this section, it seems as if the legs are still fully extended (i.e. vertical). Is this true? How did you position the legs when calculating the center of mass of each segment? Were they still vertical, and if not, how did you chose the sagittal plane angles of the lower limbs from which the CoM position would be calculated? This will affect the resultant vertical position of the CoM. It is also unclear from the section starting at line 266 if the legs were included in the body Iy calculation. Is this why the term “axial body” is used?

Agility Metrics:
The follow paragraphs are my biggest methodological concern with the paper:

The authors utilized two metrics of ‘agility’ to test their hypothesis. The first Agilityforce, was used on the basis of the human studies, (which as I pointed out were not in the bibliography, and I was not able to figure out what they actually showed), and as mentioned by the authors in the discussion (lines 593-597) looked at a different system altogether. Also, in lines 336-338, I am totally unable to understand what an agility/force correlation is. Agility is not a set physical quantity in the way that force is. Regardless, the equation for Agilityforce was Agilityforce = Ailium/I(y about the CoM). Since τ(yaw) = I(yaw)*αyaw, and τ(yaw) = r(from ilium to CoM)*F, and if Force is proportional to Area, then this can all be simplified to Agilityforce = Ailium/I(y about the CoM) = αyaw/r (from ilium to CoM). So the measurement of Agility here, is essentially equal to the maximum potential angular acceleration that can be created about the CoM, divided by the lever arm r (from CoM to Ilium), assuming that all the muscles coming off of the ilium are directed perpendicularly to the plane of the ilium. But, this is not how torques which would ‘yaw’ the body would be created. In other words, the body does not rotate about the CoM. Instead, in its simplest case, a standing position, a yaw rotation of the body would occur via lateral/medial rotators of the hip (about the hip). Therefore the most important variables would be the cross sections of the hip rotators, and their lever arm relative to the acetabulum, not relative to the CoM. From this, you could estimate maximum angular acceleration potential (I.E Agilityforce) based on rotations about the hip, not about the CoM. Furthermore, this would be done using the moment of inertia of the body about the hip rather that moment of inertia about the CoM.

The second metric, Agilitymoment is just a more direct way of calculating the torque about the CoM, but without lumping the moment arm into the equation above (i.e. the equation above boiled down to Agilityforce = r (from ilium to CoM)/ αyaw). In this new equation Agilitymoment = (r*A*20)/Iy. Since the numerator in this equation is equal to Torque about the CoM, and since torque divided by Iy is angular acceleration, this would boil down to Agilitymoment= αyaw, which essential the same as r/Agilityforce. However, the difference between the above equation for Agilitymoment= αyaw, and what was done in the paper, is that r was not directly used in the calculation, instead, a proxy of body mass was used for reasons which I don’t quite understand. The authors had calculated the CoM position in the anteroposterior direction, and assuming these specimens are fairly complete, they could have just calculated the distance between the centroid of the ilium (the resultant muscle force of iliac muscles), and the distance between the CoM. I’m not sure why the cube root of body mass was used instead. While moment arms should scale with the cube root of body mass, you can’t just substitute the cube root of body mass in an equation instead of using a moment arm. In addition, contra to the point 2) on line 354, doing this would still leave you with the allometric effects discussed on line 352. Furthermore, this approach still has the same fundamental issue as the first equation, which is that it assumes that rotations happen about the CoM, and Iy is calculated about the CoM. Rather, the cross sections of the medial and lateral rotators of the hips should be utilized, moment arms of these muscles should be utilized, and Iy of the body about the hip should be utilized.

Finally, in both approaches there is a fundamental issue that ilium area is taken as a direct measure of muscle production. In other words, this assumes that there is a single muscle, with a cross section the size of the ilium, whose fiber emanate directly perpendicular to the plane of the ilium. But, given the discussion of caudofemoralis longus, it is clear that caudofemoralis longus must travel into the tail. In which case, its fibers are not directly perpendicular to the plane of the ilium. So even if the approach taken above were appropriate (calculating torque and angular acceleration about the CoM), these muscle forces must be resolved perpendicular to the moment arm between the CoM and the resultant force vector. Not all iliac muscles which have a force vector perpendicular to the plane of the ilium. Which leads me to the question, does the caudofemoralis act as a femoral rotator in extant animals that have it?

For these reasons, I find the results of this study very difficult to interpret in any physical sense. Im not sure what ‘Agility’ here is telling us. I have a feeling that if one were to just regress iliac are on body mass, the same image that is shown in figure 4 would emerge.

Phylogenetic Methods:
The authors describe the statistical methods from lines 359-459. I think this is overkill. The method section is already necessarily long because of the modeling section. The statistics are not so complicated that they need to be several pages long. The non-phylogenetic methods are discussed first, and then the phylogenetic methods. I would like to think (though I know the older generation of paleontologists may not agree) that we are at a point where we do not have to justify using a phylogenetic method over a non-phylogenetic one. I would advocate that this section be severely reduced in length (lines 359-459). I think its sufficient to state that both OLS and PGLS regressions were performed, phylogenetic signal was detected via Pagel’s lambda, and therefore Phylogenetic ANCOVA’s were used, with perhaps a brief explanation of how they work. I would also caution that in line 373-374 phylogenetic ANCOVAs were mentioned as a way to overcome small sample sizes, which I don’t believe is true. Also, in lines 378-380, I don’t understand how this description is actually different from that of a regular ANCOVA. Both need two separate groups, you could have done the same partitioning of groups that you did in this study in a regular ANCOVA. I would suggest taking this sentence out or re-wording it. Again lines 400-417 can almost all be replaced by a single line about testing for phylogenetic signal. Lines 445-460 simply state that you used family-wise error rates to reduce the threshold for significance.

Validity of the findings

Because of the issues mentioned above, I have a hard time reconciling what the Agility metrics in this paper are actually telling us. What they are not telling us, is what the potential torque or angular acceleration about a hip could be in these dinosaurs. The authors premise the use of agility on these humans studies, but I was unable to locate them because they were not in the bibliography.

Furthermore, while I think the question that the authors are investigating is super interesting, I would have liked to see some discussion about how complicated turning is. I can understand how difficult it would be to calculate these kinds of variables, but in real life, when animals turn at high speeds, the also lean in, which changes the orientation of everything (as the authors pointed out in the intro). How would this affect the interpretations drawn in the current study?

A nit picky comment: In lines 582-583, it is said that the current study “… indicates strong relationships between morphology, agility, and invasion of a distinctive performance nice by tyrannosaurids.” The former two points are valid to say, but the third (invasion of a distinct performance niche), is not indicated by the current study; it is hypothesized on the basis of the results of the current study.

Additional comments

Due to the unfortunately ordered nature of the comment boxes in peerJ, I hope that the authors find this section first, which is that I am a fan of this paper. I think it was really a fun exercise in playing around with a way to model aspects of locomotor performance (i.e. turning performance, maneuverability) that may actually be very important in assessing an animal’s prey capture methods, the type of prey that they might be able to capture, and other aspects of fitness. Furthermore, I think the figures were all very beautiful, and I greatly applaud the use of phylogenetic ANCOVAs utilizing Smaers and Rohlf (2016). I think that method was a perfect way to test the hypotheses that the authors sought to investigate. I sincerely hope that my commentary was helpful, and that the authors find ways to address what I perceived as some of the issues with the study, and that it will ultimately be published. I think that it will be of interest to many fields of paleontology and biomechanics.

Following below is a series of more minor comments and suggestions for changes in wording.

Lines 65-67: Sentence: “Here we quantify biomechanical evidence that …”. This paper is not quantifying biomechanical evidence, this paper is presenting a biomechanical model, which suggests that tyrannosaurids could turn more rapidly, I would suggest rewording this.

Lines 69: Suggest removing “locomotor” and just leaving “muscle force”. Moments about joints don’t care if the muscle function is “locomotor” or anything else.

Line 101: “a proxy for muscle cross section area and force” I would change to “… cross sectional area and maximal force production”

Line 120-121: I’m not sure what the line “Most other taxa were allosaurioids to enforce some phylogenetic consistency for other large theropods” means. Do you mean that you just need a sufficient number of outgroup taxa to tyrannosaurids?

Lines 123-125: These sentence refer to taxa that were omitted and I think is unnecessary.

The two sentences on lines 140-146 are basically a repeat of one another.

Lines 178-179: I would suggest this edit: …. Tyrannosaurids would lower their relative MMI as compared to the larger forelimbs….

Line 231 : Take out the word “Furthermore”

Lines 234 -235, I would add something like: “…than those previous estimated by the method herein employed for dinosaurs”

Lines 236: Suggest replacing “to obtain the upper end of the range for tail” by “to obtain an upper estimate of”. It could always have been higher in extinct taxa.

Line 248: What does “axial bodies” mean?

Line 294-295: How can you be sure that it’s a good estimation? Change wording to “an estimation”. Also Im not sure if the term CFL was ever defined.

Line 297: Is it a leg, or femoral retractor?

Line 350: I would like to see a citation for the 20 N/cm value.

Line 470 and elsewhere: NURBS is never spelled out.

Lines 486-488: This is an oddly worded sentence and I couldn’t quite get the gist of it.

Lines 531-533 are unnecessary, and it shouldn’t be expected that the R value is the same between the two regressions. The takeaway is it is high in both.

Lines 534-535: I don’t quite understand this sentence.

Lines 536-538: I see one blue dot that does fall within the PI on Figs 5 and 6, which point is this?

Lines 554: Figs 3 and 4 should be 5 and 6.

Lines 557: The method you used isn’t novel, it’s recently published.

Lines 593-596: Again, here is reference to the human study, which is not thoroughly described or in the bibliography.

Lines 617-618. Take this sentence out, the timing of who did what first is irrelevant.

Line 639-640: Is this fact relevant to the current paper?

Lines 641-645: It is good that you tried this multiple ways, and have that data in the appendix. That is where figure 3 should be as well. Furthermore, the final sentence in this paragraph is perplexing. If the simpler method is adequate, why do you only mention it at the end of the discussion?

Lines 651-653: This sentence is an assumption that we all hope is true. I would reword it as such.

Lines 700-702: This is the second time this is mentioned, but isn’t it actually disproven in line 706-707? Unless Dakotaraptor at 374 kg is smaller than a wolf.

Lines 709-710: I applaud the authors for having a very illustrative form of writing, but I have no idea how many kilograms a lion is, or a jackal is, or an African wild dog is. I would say just stick with the previous sentence on the 17-24 x bigger. That I can visualize.

Column z in Table 3 is unnecessary.

In Table 4, I would suggest changing the last three columns to percentages instead of 1.xxx.

·

Basic reporting

Basic Reporting is sound.

Experimental design

The Experimental Design is excellent. My primary concern while reading the body of the manuscript was the potential for sensitivity to body mass distribution estimation and modeling bias in the proportions. Both of these have been well addressed in the appendices through cross-method comparisons and sensitivity analysis. My only comment would be that some mention of these additional results earlier in the manuscript may be warranted, so that readers are not distracted throughout by the worry that these issues have not been addressed (until the end).

Validity of the findings

As with all modeling studies, the validity of the findings pivots entirely on the strength and precision of the initial model. The design is sound, and therefore the confidence in the results is valid. The connection to macroevolutionary trends is speculative, but it is a level of speculation I found quite reasonable (after all, it really is a bit weird that tyrannosaurs have so much of the Late Cretaceous predator ecology to themselves). It would be interesting to consider the potential impacts of pre-existing adaptations in the ancestral state for tyrannosaurs that might have predisposed the clade for exceptional locomotor performance (i.e. is there a phylogenetic reason why tyrannosaurs ended up with a high-agility, high-speed body plan that scaled well to large size?)

Additional comments

My one gripe with the manuscript, which is easily fixed, is the issue of maneuverability vs agility. Technically, the former is typically used in the technical literature to mean minimum radius of turn, while the latter refers to maximum rate of turn. Based upon your parameters, I think the parameter you're really estimating is relative agility, so it makes sense that this is what you mostly use. However, there are places where maneuverability is referenced, instead.

I realize that there are more informal usages for these terms, and that is likely why you use them both. However, I think it is important that both terms are defined early on (even if that means just clearly indicating that you are using them in a general, interchangeable manner). I also think it's worth indicating what components of turning performance your model is likely to estimate (i.e. is your model mostly predicting rate of turn performance, radius of turn performance, or some combination of the two that you don't expect can be separated? - as noted above, I suspect it is mostly rate of turn)


One very minor point (address at your discretion):
“Relative maximal muscle force is straightforward to estimate directly and consistently from fossil evidence, compared to musculoskeletal moment arms that vary with posture, or physiologically variable factors such as muscle power (Young et al. 2002)”

This is true, but the catch is that for measures which intrinsically involve a rate (such as agility - i.e. rate of turn), power is likely a key variable. Of course, maximal relative muscle force will still be a strong predictor in most cases (since physiological muscle differences will only vary so much, especially when comparing animals with similar mass and ecology), and a valid way of dealing with the force vs power problem in the context of turning performance is to use a comparative approach - which is exactly what you’ve done. I do think a brief additional note about this additional advantage of the comparative approach is warranted. This paper will draw a fair bit of interest from readers who are new to biomechanics (because tyrannosaurs), and they might not all recognize the full range of problems you are solving by using a comparative approach with relative metrics.

---

## Round 0.2 · Minor Revisions

Thank you for your close attention to the previous round of comments. After a second review, just a handful of comparatively minor issues remain. Please address them in your revision, either by incorporation into the manuscript or with a rebuttal in the response letter.

Reviewer 1 ·

Basic reporting

The authors have made a number of changes which I believe have improved the overall manuscript. In particular, they have added more details about the assumptions of their hypotheses, and added various other methodological details which I find makes the manuscript easier to read and interpret. I appreciate the authors’ diligence to addressing my previous comments! I think this is an interesting study, but there are a number of issues that should still be addressed, which I have no doubt the authors will be capable of addressing. There is one larger issue however, which I detail in the experimental design below.

Experimental design

My larger comment pertains to the underlying assumption/link between muscle force and agility. After consulting the Peterson, Thomas and Weiss papers, I would strongly suggest the authors be a more careful about their wording associating muscle force to agility in humans. The authors are using this as the justification for looking at muscle force (i.e. cross sectional are) as a predictor of agility in dinosaurs. But the cited studies (Peterson et al., 2006; Thomas et al., 2009; Weiss et al., 2010) don’t quite show what the authors claim they show.

All three of these studies are human studies correlating performance on agility tests with either other exercises (squats) or different training regimes. In these contexts it’s not surprising that people that can squat heavier amounts, or train for several weeks, are better at agility tests. But importantly, throughout this manuscript, the authors explicitly make the statements (or variants of the statement): “Muscle force…. is a direct correlate of agility in humans” (Line 40 and elsewhere in the paper). However, none of these studies measure muscle force, nor maximum muscle force, nor any prediction of any muscle forces (despite the somewhat misleading title of Peterson et al., 2006). This statement is not supported by the cited studies.

I see in the authors view the best evidence probably comes from Peterson et al 2006, but this is simply correlating T-test agility scores with 1RM max squat load. While its assumed this is related to muscle mass, Thomas et al., (2009) actually state that the improvements in agility they measured were likely not muscular: “The adaptations to both forms of training are likely to be neural because these predominate in the early stages of strength and power training (29) and have been shown to be the main adaptation to plyometric exercise (15)”. While of course at some level, if agility is correlated with max squat load, and presumably max squat load is going to be dependent on muscle mass, then muscle mass will be correlated with agility. But how much so, and of what muscles? The cited studies don’t address this, but as they are cited by the authors, it gives the sense that they do, even though this is dispelled later in the discussion (lines 734). This tends to permeate in language in other sections as well, such as line 463 (“We developed two indices of relative agility for theropods: Agilityforce based on agility/force correlations in humans (Peterson et al. 2006, Thomas et al. 2009, Weiss et al. 2010)”. But I don’t see how the metric here is in any way related to the cited papers, nor do the cited papers show that muscle force is correlated with agility (again because they did not measure or estimate muscle force). I think this kind of language gives a false sense of confidence about the relationship between muscle mass and agility that is not supported with the citations included. Now, I do not criticize the authors for developing their own metric, nor do I think this obviates the point of the paper; I do think though that the authors should more clearly delineate what the cited studies actual show, and how they are using some of these assumed relationships as a springboard to develop their own metrics, that hopefully tell us something about turning ability.

As a final thought on this topic, in this manuscript, agility is used as an indicator of turning performance; in Peterson et al., (2006), they measure agility as a T-test, which actually doesn’t involve any turns.

I would strongly encourage that the authors remove or edit the sentence “Relative (not absolute) maximal muscle force is straightforward to estimate directly and consistently from fossil evidence, compared to musculoskeletal moment arms that vary continuously with posture in three dimensions, or physiologically variable factors such as muscle power (Young et al. 2002).” First of all, I don’t think that this is true. Yes, moment arms vary with posture, but muscle force is still harder. You need PCSA, which in and of itself involves fiber length, muscle volume, and pennation angle, and only one of these (muscle volume), can be approximated in a straightforward way from a fossil. Furthermore they change with age and health and other factors. I also don’t quite understand the second qualm about muscle power, nor does Young (2002) seem to have anything to do with muscle moment arms. I understand the logic behind wanting to use muscle cross sectional area for the current study; I don’t think it is necessarily to criticize muscle moment arms (criticism with which I don’t agree) in order to get that point across. If muscle force is straightforward and easy to measure in therapods, then great, I think its sufficient to leave it at that.

Validity of the findings

I would take an additional look at the PGLS/ANCOVA results that you show in Fig. 4–5. PGLS sometimes makes the regression lines look weird when plotted on the raw data, but it seems a bit weird that for the ‘Therapod’ line, all but one point is above the regression. I know I’ve played with datasets where the slope looks funky when plotting the PGLS line on the raw data, but I haven’t seen one where nearly all the points have one-sided residuals. I would just double check this. Similarly, I’m not sure what the statement on lines 664–666 means about ‘under-predicting’ agility.

Additional comments

One burning question I still have stems from the introduction; that tyrannosaurids have “tall ilia for leg muscle attachment”. If you were to create two separate plots similar to Fig 3, for both Ilium Area, and Iy separately, do they both show the same trend, or is this all just a matter of pelvis size?

I know I mentioned this in my first review, and the authors have added a line in the paper, but you are using two abbreviations for moment of inertia (MMI and I). Mass moment of inertia is commonly abbreviated I, and it distracting to have two different abbreviations for the same thing in one paper. Either use I, or MMI, but don’t switch back and forth.

Line 111: Divided ilium area by what?

Line 151: Is the COM set to be directly above the foot? Assuming that this was during locomotion, the COM during single stance would probably be medial to the stance leg. Just something to think about, though I doubt it would affect the results.

Line 311: Though they are technically results, it might help to have the inter-observer error results (or %’s) at the end of the paragraph on line 311, so we don’t have to be in suspense!

Line 398–404: I’m not sure what these examples provide; they are stuck in a section describing the accuracy with which muscle scars preserve actual muscle size, but neither of the examples seem to directly address this. Perhaps re-word or remove this section.

Lines 474–484: There are a few things that could be done to make the units more consistent. In Equation 13, is Iy measured in kg*m2? And if so are you mixing cm and m measurements? Similarly in equation 14 are cm and m units being mixed?

I would move the subsection “Statistical comparison of group differences: phylogenetic ANCOVA” to after the section “Determining strength of phylogenetic signal and appropriateness of phylogenetic regression”, or combine it with the section “Phylogenetically informed analyses”.

Check for consistency in the numbers of significant digits in the tables and throughout the text.

Line 711: “uniquely superior” is a pretty subjective statement that don’t really mean anything. Can you be more specific?

Lines 715–717: I would suggest changing this sentence. It is a bit of a stretch to claim that you found a strong relationship between something and “distinct performance capacity by tyrannosaurids”. You are inferring a performance difference given a metric you’ve created that you believe to be informative.

The discussion would be a good place to further expand on what specific is driving the difference in ‘agility’. Is it all Ilium area, or a bunch of factors?

Minor comments:

Line 78: Perhaps after the line ending “…(=rotational inertia).”, preface the sentence that you are about to talk about turning strategies aside from simply increase torque.

Line 87–88: “When limbs are planted on the ground, the body can pivot with locomotor muscle alone”. I’m not entirely sure what this sentence is trying to say; is it separating the situation of both legs planted on the ground? Also, I would suggest changing ‘locomotor muscle’ to ‘muscle force’.

Check the number of significant digits in tables and text for consistency.

·

Basic reporting

No comment

Experimental design

No comment

Validity of the findings

No comments

Additional comments

I have (finally) read through the new version of the manuscript in detail. All of my prior concerns were handled, and I note that significant improvements were made in response to the comments of the second reviewer (who caught some potential issues that I missed and/or was less concerned by). I did not find any errors or new problems introduced by these additions and revisions. I note that additional equations and tables were now required to explain the new analyses. These are warranted and necessary in the new version. I also wish to apologize to the authors for the delay in my review. You deserve to know that it was my review that held this up, and the delays had nothing to do with the quality of the manuscript or my interest in the work.

---

## Round 0.3 · accepted · Accept

Thank you for your close attention to the comments from the reviewers. From the scientific end of things, I believe the paper is ready for publication.

I note two items that will need to be addressed in production (but do not, in my opinion, require a formal revision stage):

1) The manuscript lacks an acknowledgments section; is this intentional or accidental?

2) Some of the figures may require additional permissions for use in this paper under a CC-BY license--please confirm that you have appropriate permissions to use all artwork that is adapted from other sources (e.g., the ones modified from Paul; I note that Hartman is a co-author, so I am assuming those are all OK).

#